# Apelin-VEGF-C mRNA delivery as therapeutic for the treatment of secondary lymphedema

Justine Creff [1,7], Asalaa Lamaa[1,7], Emeline Benuzzi[1], Elisa Balzan [1], Francoise Pujol[1], Tangra Draia-Nicolau [1], Manon Nougué[1], Lena Verdu [1], Florent Morfoisse [1], Eric Lacazette [1], Philippe Valet[2], Benoit Chaput[3], Fabian Gross [4], Regis Gayon[5], Pascale Bouillé[5], Julie Malloizel-Delaunay[6], Alessandra Bura-Rivière [6], Anne-Catherine Prats [1] & Barbara Garmy-Susini [1]✉

## Abstract

Secondary lymphedema (LD) corresponds to a severe lymphatic dysfunction leading to the accumulation of fluid and fibrotic adipose tissue in a limb. Here, we identified apelin (APLN) as a powerful molecule for regenerating lymphatic function in LD. We identified the loss of APLN expression in the lymphedematous arm compared to the normal arm in patients. The role of APLN in LD was confirmed in APLN knockout mice, in which LD is increased and associated with fibrosis and dermal backflow. This was reversed by intradermal injection of APLN-lentivectors. Mechanistically, APLN stimulates lymphatic endothelial cell gene expression and induces the binding of E2F8 transcription factor to the promoter of CCBE1 that controls VEGF-C processing. In addition, APLN induces Akt and eNOS pathways to stimulate lymphatic collector pumping. Our results show that APLN represents a novel partner for VEGF-C to restore lymphatic function in both initial and collecting vessels. As LD appears after cancer treatment, we validated the APLN-VEGF-C combination using a novel class of non-integrative RNA delivery LentiFlash® vector that will be evaluated for phase I/IIa clinical trial.

**Keywords** lymphedema; Apelin; VEGF-C; mRNA; Collector
**Subject Categories** Cancer; Vascular Biology & Angiogenesis

## Introduction

Lymphedema (LD) is a multifactorial condition that substantially affects the quality of life of patients (Greene et al, 2012; Hoffner et al, 2018). It is characterized by a failure of the lymph transport back to the blood circulation due to a lymphatic dysfunction that occurs after genetic mutation (primary LD) or arises after cancer treatments (chemotherapy and radiation therapy) in western countries (secondary LD) (Mortimer and Rockson, 2014). The major hallmark of LD is the development of fibrosis into the skin and the adipose tissue (AT) as lymphostatic fibrosis defines the stage of LD from reversible to elephantiasis stages (Mortimer and Rockson, 2014). Once fibrosis develops, tissues become denser leading to a lymphatic vessel obstruction that worsens LD. Importantly, fibrosis also affects collecting lymphatic pumping and increases limb swelling (Baik et al, 2022; Kataru et al, 2019). A large number of cytokines and peptides are known to selectively improve adipocyte metabolism, endothelial function or tissue fibrosis. However, the bioactive peptide apelin (APLN) combines these beneficial effects as a whole. APLN is the endogenous ligand of the G-protein-coupled receptor APJ, and is a critical actor of the fibrosis protection in many organs including the heart, lung, and kidney (Huang et al, 2016; Pope et al, 2012). The first evidence of the link between APLN and lymphatic vasculature were identified in a tumoral environment as APLN stimulates both hemangio- and lymphangiogenesis (Berta et al, 2014). In particular, APLN stimulates NO production via PI3K/Akt signaling in blood endothelial cells (Busch et al, 2015). Also, our group found that APLN was able to restore the lymphatic shape of precollecting vessels after cardiac ischemia, suggesting that APLN may represent a good candidate to restore the lymphatic shape in injured tissues (Tatin et al, 2017).

Based on data accumulated throughout 20 years of clinical trials, gene therapy has an acceptable safety profile for vascular diseases (Gupta et al, 2009). However, more rigorous phase II and phase III clinical trials have failed to demonstrate that growth factors administered as single-agent gene therapies are beneficial in patients with cardiovascular diseases (Gupta et al, 2009). Also, secondary LD related to cancer treatment is an ongoing challenge, which forces therapies to avoid any effect on cancer recurrence. Recent Covid19 vaccines has demonstrated that mRNA can be used as a way of delivering therapeutic proteins. But compared to

[1]I2MC, Université de Toulouse, Inserm UMR 1297, UT3, Toulouse, France. [2]Institut RESTORE, UMR 1301-INSERM, 5070-CNRS, Université Paul Sabatier, Université de Toulouse, Toulouse, France. [3]Department of Plastic Surgery, University of Toulouse III Paul Sabatier, Toulouse, France. [4]Biotherapy Module of Clinical Investigation Center (CIC 1436), University Hospital of Toulouse, 31059 Toulouse, France. [5]Flash Therapeutics, Toulouse, France. [6]Service de Médecine Vasculaire, Centre Hospitalier Universitaire de Toulouse, Toulouse, France. [7]These authors contributed equally: Justine Creff, Asalaa Lamaa. ✉E-mail: barbara.garmy-susini@inserm.fr

vaccines, mRNA therapeutics requires a 300–4000-fold higher level of protein expression to reach a therapeutic effect when they are delivered through lipid nanoparticles due to a poor entry efficiency into target cells and synthetic RNA stability (Rybakova et al, 2019). To overcome these limits and enhance both duration and protein expression level in vivo, we propose to use a biological RNA delivery technology, LentiFlash®, to transfer two different mRNAs into LD tissues. Therefore, we propose to combine APLN to VEGF-C, the major lymphangiogenic factor and the only one molecule that has been evaluated in a clinical trial for secondary LD treatment (Hartiala et al, 2020a). VEGF-C biological effect is enhanced by the collagen- and calcium-binding EGF domains 1 (CCBE1) along with a disintegrin and metalloproteinase with thrombospondin motifs-3 (ADAMTS3) protease (Jha et al, 2017). CCBE1 is a secreted molecule involved in lymphatic vasculature development and mutations in *CCBE1* were identified in patients with Hennekam syndrome, a rare autosomal recessive disorder of lymphatic development leading to primary LD (Bonet et al, 2022; Hogan et al, 2009; Kunnapuu et al, 2021; Van Balkom et al, 2002).

Here, we found a significant decrease of APLN in biopsies from patients who develop LD after breast cancer. We showed that APLN stimulated gene expression in LEC through E2F8, a part of E2F family of transcription factors that is an important regulator of lymphangiogenesis in zebrafish and mice after binding to CCBE1 and Flt4 promoters (Kunnapuu et al, 2021; Lammens et al, 2009; Logan et al, 2005; Weijts et al, 2013). We identified APLN as a crucial player in the NO-induced lymphatic pumping by stimulating eNOS phosphorylation in LEC. Thus, we performed a multiple gene therapy by combining VEGF-C with APLN to obtain a synergistic effect necessary to restore the lymphatic function. Our study provided evidence for the use of APLN-VEGF-C combination in patients who developed LD after cancer treatments. Here, we present the preclinical study using APLN-VEGF-C mRNA delivery vectors that will be used in a phase I/II gene therapy clinical trial for the treatment of patients who developed LD after breast cancer.

# Results

## LD exhibits increased lymphatic capillary diameter and poor collecting drainage

Secondary LD occurs months, sometimes years after cancer treatment, suggesting that this pathology is not only a side effect of the surgery. Lymphofluoroscopy of women who developed LD after breast cancer showed hypervascularized dermis with tortuous lymphatic capillaries associated with a strong desmoplastic reaction and dermal backflow (Fig. 1A). Lymphoscintigraphy is the primary imaging modality used to assess lymphatic system dysfunction. It has been considered the criterion standard for decades (Munn and Padera, 2014; Szuba et al, 2003). Lymphoscintigraphy revealed a severe decrease in lymphatic collecting vessels detection and axillary lymph node perfusion after radiotracer injection (Fig. 1B). Histological analysis showed an increase in dilated lymphatic vessels in the skin (Fig. 1C–E). This was associated with a strong fibrosis development in both skin (Fig. 1F) and AT (Fig. 1G). Surprisingly, no major difference in genes involved in lymphangiogenic factor maturation was observed except for CCBE1 which was

significantly downregulated in LD (Fig. 1H). As LD is characterized by a strong accumulation of fibrotic AT in the limb, we also evaluated adipokines expression in the lymphedematous AT compared to the normal arm (Fig. 1I). We found a significant decrease in APLN expression in LD, whereas no difference was observed for other adipokines (Fig. 1I).

## Impaired lymphatic healing in APLN knockout mice

APLN was previously described by our group to improve lymphatic vessel normalization in the heart after cardiac ischemia (Tatin et al, 2017). To investigate the role of APLN in secondary LD, we used a LD mouse model previously developed in the laboratory (Morfoisse et al, 2018). We performed second mammary gland mastectomy associated with axillary and brachial lymphadenectomy on the upper left limb in APLN-KO mice (Fig. 2A). Using this model, reproducible LD developed after 2 weeks to progressively return to normal after 4 weeks. In APLN-KO mice, LD was significantly maintained after 4 weeks, showing the failure to restore the lymphatic function (Fig. 2A). Lymphatic capillaries were next investigated using lymphangiography after footpad injection of FITC-Dextran (Fig. 2B). We observed a strong dermal backflow in APLN-KO mice 4 weeks after surgery (Fig. 2B). Histological analysis of the lymphatic showed no difference in lymphatic basal density between WT and APLN-KO mice (Fig. 2C,D). In contrast, in LD, the skin lymphangiogenesis was significantly reduced in APLN-KO mice compared to WT mice (Fig. 2C,D). This was associated with an increase in skin fibrosis in both WT and APLN-KO mice as shown using Masson's trichrome staining (Fig. 2E,F). As LD results in an accumulation of collagen fibers, one of the hallmarks of fibrosis development, we performed skin analysis by second harmonic generation (SHG) imaging (Fig. 2G,H). Interestingly, the accumulation of collagen fibers increased in LD in APLN-KO mice compared to WT mice (Fig. 2H).

## APLN possesses regenerative function on lymphatic vessels in LD

We next investigated the role of APLN on LD-bearing wild-type mice (Fig. 3). In mice skin, APLN receptor APJ was mainly found on LEC and fibroblasts (Fig. EV1A). As expected, APLN mRNA expression was significantly reduced in LD limb compared to the normal limb, as observed in human tissue samples (Fig. EV1B). To evaluate the effect of APLN on lymphatic healing, mice received an intradermal injection of APLN-expressing lentivector (LV-APLN) in the lymphedematous limb. Remarkably LD was significantly reduced in APLN-treated mice (Fig. 3A). The Level of circulating APLN was verified by ELISA dosage on mouse plasma showing an increase in plasmatic APLN concentration in LV-APLN-treated mice (Fig. 3B). Lymphatic collecting drainage was next investigated using lymphangiography (Fig. 3C). We observed that secondary LD induced a pathological remodeling of lymphatic vessels with disorganized and abnormal vessel morphology and increased number of branching regarding the control limb. Lymphatic leakage (dermal backflow) was also observed, revealing a dysfunction in the superficial overloaded capillary network due to the lack of deeper collector pumping (Fig. 3C). On the contrary, APLN-treated mice displayed an improvement in the lymphatic shape

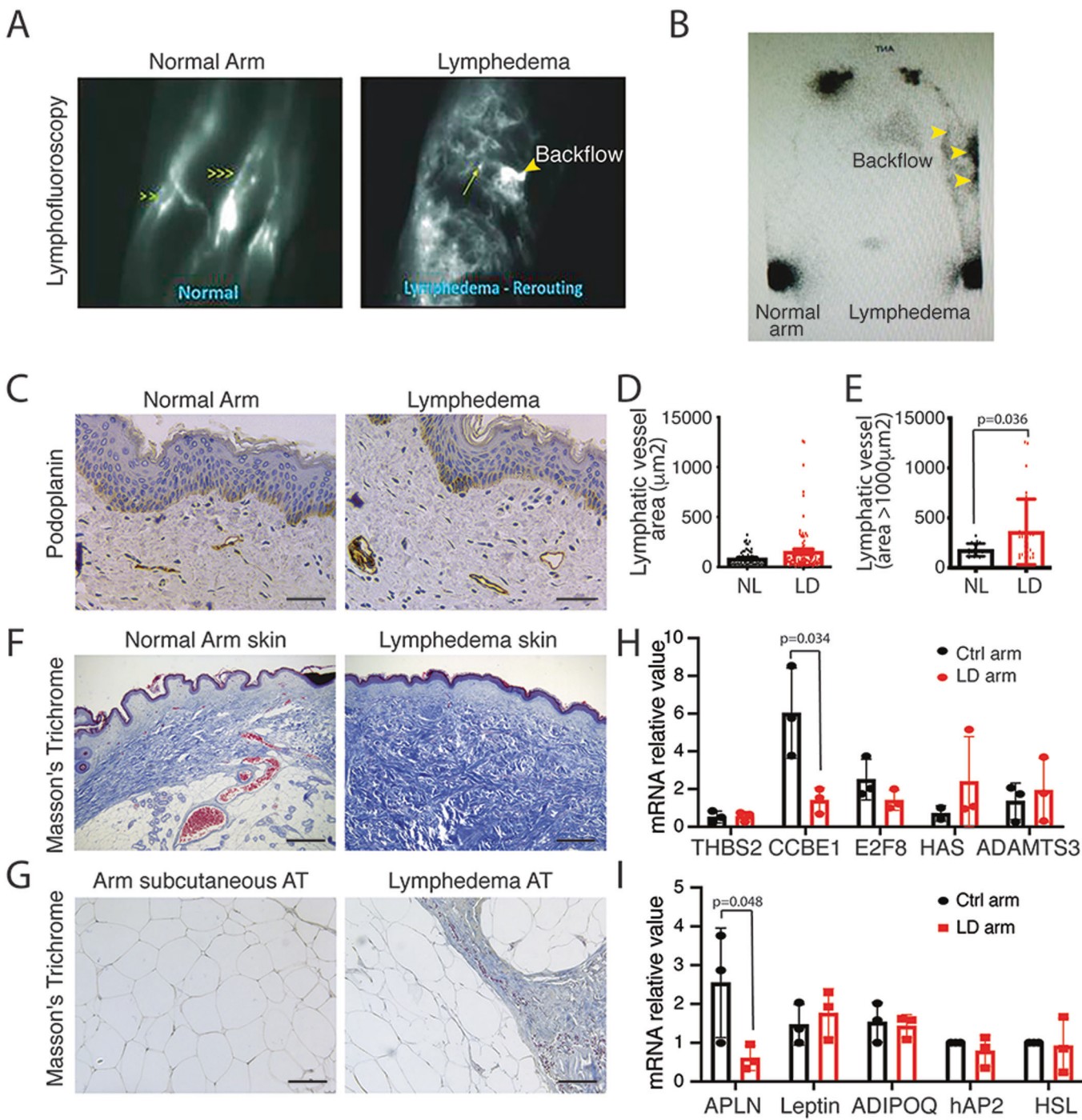

**Figure 1. Reduced APLN expression in human LD.**

(A) Lymphofluoroscopy of the upper limb LD shows dermal lymph backflow associated with hypervascularization of tortuous initial lymphatics (right panel) compared to normal arm (left panel). (B) Lymphoscintigraphy of woman who developed LD after breast cancer shows reduced but lasting lymphatic drainage and lymph node perfusion. (C) Immunodetection of the lymphatic networks in the LD skin shows dilated lymphatic vessels (scale bar: 50 μm) (n = 8). (D) Quantification of the lymphatic vessel diameters. Data represent mean ± SEM (n = 8). (E) Quantification of the dermis lymphatic density. Data represent mean ± SEM of three biological replicates (unpaired t test; *P < 0.05). (F) Masson's trichrome staining of the human lymphedematous skin shows strong fibrosis (scale bar: 50 μm) (n = 8). (G) Masson's trichrome staining of the human LD subcutaneous AT shows fibrosis (scale bar: 50 μm) (n = 8). (H) Quantitative RT-PCR analysis of the genes involved in fibrosis and VEGF-C maturation in dermolipectomy samples from patients with LD. Data represent mean ± SEM of three biological replicates (paired t test; P = O.O34). (I) Quantitative RT-PCR analysis of the adipokines in dermolipectomy samples from patients with LD. Data represent mean ± SEM of three biological replicates (paired t test; P = 0.058). Source data are available online for this figure.

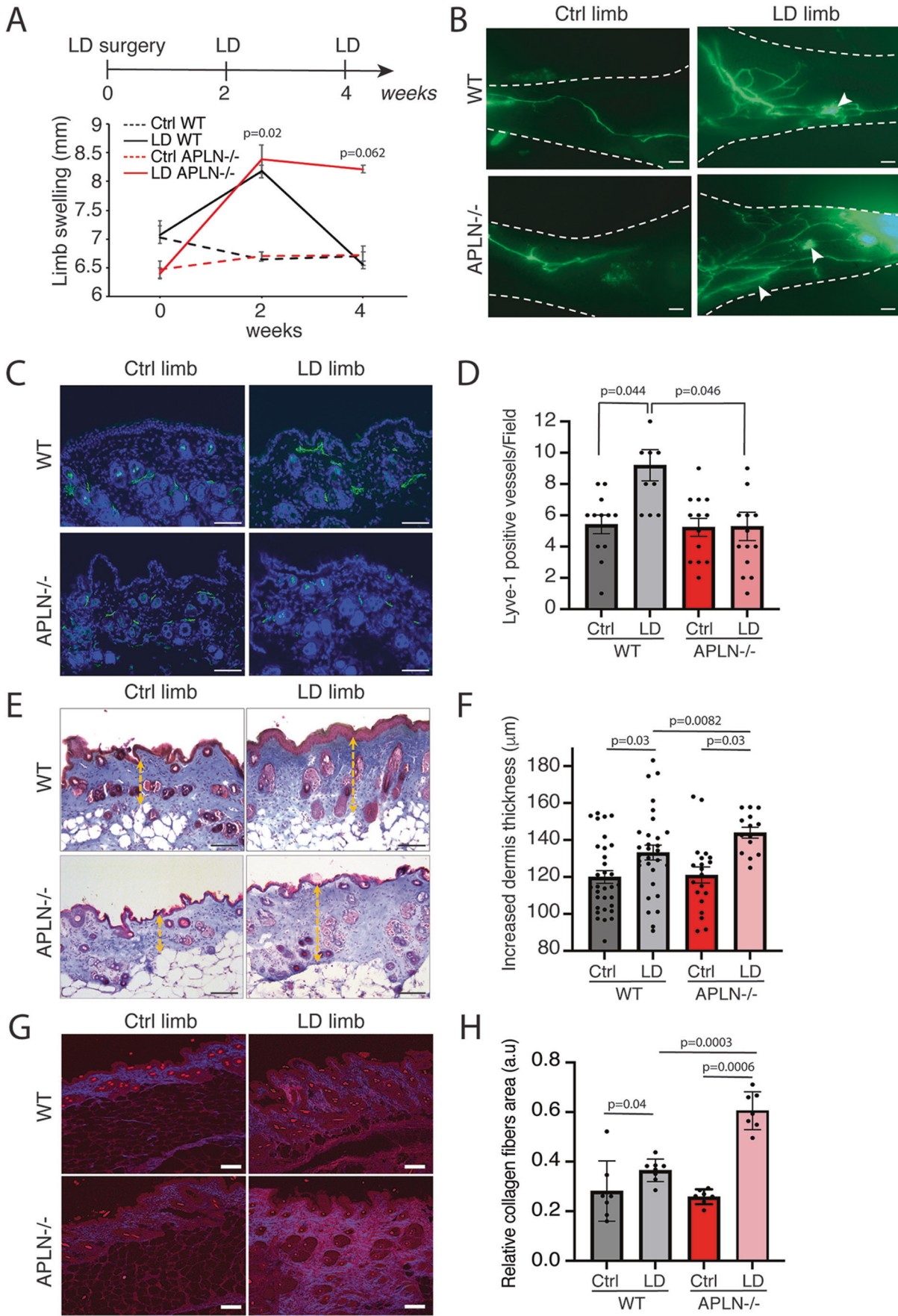

**Figure 2.   LD increases in APLN-KO mice.**

(A) Schematic of the experimental design of secondary LD mice model. Quantification of proximal limb swelling 2 and 4 weeks after surgery on control limb, LD limb from APLN-KO mice and control littermates. Data represent mean ± SEM ($n = 9$) (two-way ANOVA). (B) Lymphangiography reveals dermal backflow (white arrows) and pathological remodeling of lymphatic vessel after LD in APLN-KO mice 4 weeks post surgery (scale bar: 1 mm). (C) Lyve1 immunodetection of the skin lymphangiogenesis in APLN-KO mice (scale bar: 50 µm). (D) Quantification of lymphangiogenesis in the skin from APLN-KO mice. Data represent mean ± SEM ($n = 10$–14 mice) (one-way ANOVA). (E) Masson's trichrome staining of LD in APLN-KO mice (scale bar: 50 µm). (F) Quantification of dermis fibrosis in APLN-KO mice 4 weeks post surgery. Data represent mean ± SEM ($n = 5$) (one-way ANOVA). (G) SHG signal from deep, collagen-rich layer within dermis (scale bar: 50 µm). (H) Quantification of the relative collagen area. Data represent mean ± SEM ($n = 7$–8) (one-way ANOVA). Source data are available online for this figure.

with normalized morphology and decreased number of vessels branching. Importantly, we did not observe dermal backflow in APLN-treated mice, suggesting an improvement of lymphatic function (Fig. 3C). Using Masson's trichrome coloration, we observed an increase of the dermis thickness in LD limb consistent with the development of fibrosis (Fig. 3D). In APLN-treated mice, we did not observe any thickening of the dermis (Fig. 3D,E) reflecting an improvement of LD pathology. Interestingly, we observed an increase in circulating VEGF-C, the major lymphangiogenic factor, after LV-APLN treatment suggesting that APLN may in part regulate VEGF-C protein synthesis (Fig. 3F). Positive control was performed using VEGF-C-expressing lentivector (LV-VEGF-C) (Fig. 3F). The effect of APLN was also evaluated on collagen deposition using SHG (Fig. 3G,H). We found a significant reduction of fibrosis in APLN-treated mice compared to control (Fig. 3H). The number of blood vessels was assessed using CD31 immunostaining (Fig. EV1E,F). As expected, we did not find changes in the number of CD31-positive vessels in this model of LD (Fig. EV1F) (Morfoisse et al, 2018). However, as previously described in the literature, treatment with APLN lentivector led to an increase in angiogenesis (Fig. EV1F) and blood vessel permeability (Fig. EV1G,H) (Wysocka et al, 2018). In parallel, lymphangiogenesis was evaluated using Lyve1 immunostaining on skin sections (Fig. 3I–K). In line with lymphangiography results, an increased number of Lyve1-positive vessels was observed in LD limb in comparison to the control limb without a significant difference when comparing control to LV-APLN-treated mice (Fig. 3J). In contrast, we found that APLN promoted significant dilatation of lymphatic vessels (Fig. 3K). Overall, our results indicate that APLN has a beneficial role on secondary LD by acting on lymphatic vessel plasticity and dilatation.

## APLN controls LEC gene expression

In order to investigate molecular mechanisms regulated by APLN in LEC, we performed a global transcriptomic analysis on LEC stimulated 24 h with conditioned media containing APLN or conditioned media obtained from control NIH3T3 (Fig. 4A–E). Differential DESeq analysis revealed that 217 genes were deregulated (P.adj<0.05 and Log2 fold change < −0.5 or >0.5) with 94 genes upregulated and 123 genes downregulated (Fig. 4A). Top 30 down- or upregulated are displayed on heatmaps (Fig. 4B) and complete lists are given in Tables 1 and 2. Gene ontology (GO) analysis of downregulated genes revealed that no biological process is significantly affected in APLN-treated HDLEC. In contrast, GO analysis for biological process revealed that upregulated genes were enriched (FDR < 0.05) for terms related to extracellular matrix (ECM) remodeling and signalization (Fig. 4C) including COL1A,

FBN, ADAMTS2, and CCBE1 (Fig. 4B,C). However, most of this gene induction was not validated by RT-qPCR on HDLEC (Fig. EV2A–C) except for CCBE1 whose induction was strongly confirmed (Fig. 4D). CCBE1 protein is required for the activation of VEGF-C along with the ADAMTS3 protease by enhancing the cleavage activity of ADAMTS3 and by facilitating the maturation of VEGF-C into its bioactive form. To investigate the effect of CCBE1 on VEGF-C receptor activation, we performed the knockdown of CCBE1 in LEC using siRNA (Fig. EV3A–D). Then, cells were stimulated by APLN, and western blot analysis of P-VEGFR3 was performed (Fig. EV3). We found that the knockdown of CCBE1 in LEC decreases the amount of VEGFR3 protein. This was associated with a slight, but significant decrease of VEGFR3 phosphorylation in the presence of APLN. (Fig. EV3A–D). Interestingly, APLN also stimulated the expression of E2F8, the CCBE1 transcription factor (Fig. 4E). We then postulated that APLN could participate to VEGF-C maturation by increasing E2F8 DNA binding on CCBE1 promoter. To answer this question, we performed chromatin immunoprecipitation (ChIP) of E2F8 in APLN-overexpressing HDLEC (Fig. 4F–I). We found that APLN significantly increased E2F8 binding to CCBE1 promoter (Fig. 4G). Interestingly, APLN also induced E2F8 binding to E2F1 transcription factor promoter, suggesting a role in other biological functions (Fig. 4H) (Wells et al, 2002). However, no binding on FLT4 was found (Fig. 4I). Altogether, these data indicated that APLN regulates HDLEC gene expression.

## APLN stimulates LEC function through Akt/eNOS signaling

Next, we investigated the effect of APLN on HDLEC at the cellular level. APLN is known to activate Erk and Akt signaling in vitro in human dermal LEC (HDLEC) (Berta et al, 2014; Kim et al, 2014). In line with the vasodilation phenotype (Fig. 3), we postulated that APLN beneficial effect on LD is in part mediated by AKT/eNOS pathway. To this end, we stimulated HDLEC with conditioned media obtained from LV-APLN transduced NIH3T3 previously depleted for VEGF-C. APLN synthesis was validated by RT-qPCR on NIH3T3 (Fig. 5A) and by ELISA (Fig. 5B). Stimulation of HDLEC by conditioned media was confirmed by evaluating AKT et ERK pathway during 24 h time course. Medium containing VEGF-C was used as positive control (Fig. 5C). Interestingly, HDLEC responded similarly to VEGF-C and APLN after 30 min as we observed a strong activation of AKT and ERK (Fig. 5C,D). This was associated with an activation of HDLEC migration by APLN (Fig. 5E,F) and an improvement of the actin cytoskeleton remodeling showing cortical actin rim compared to stress fibers observed in negative control (Fig. 5G). However, no effect was

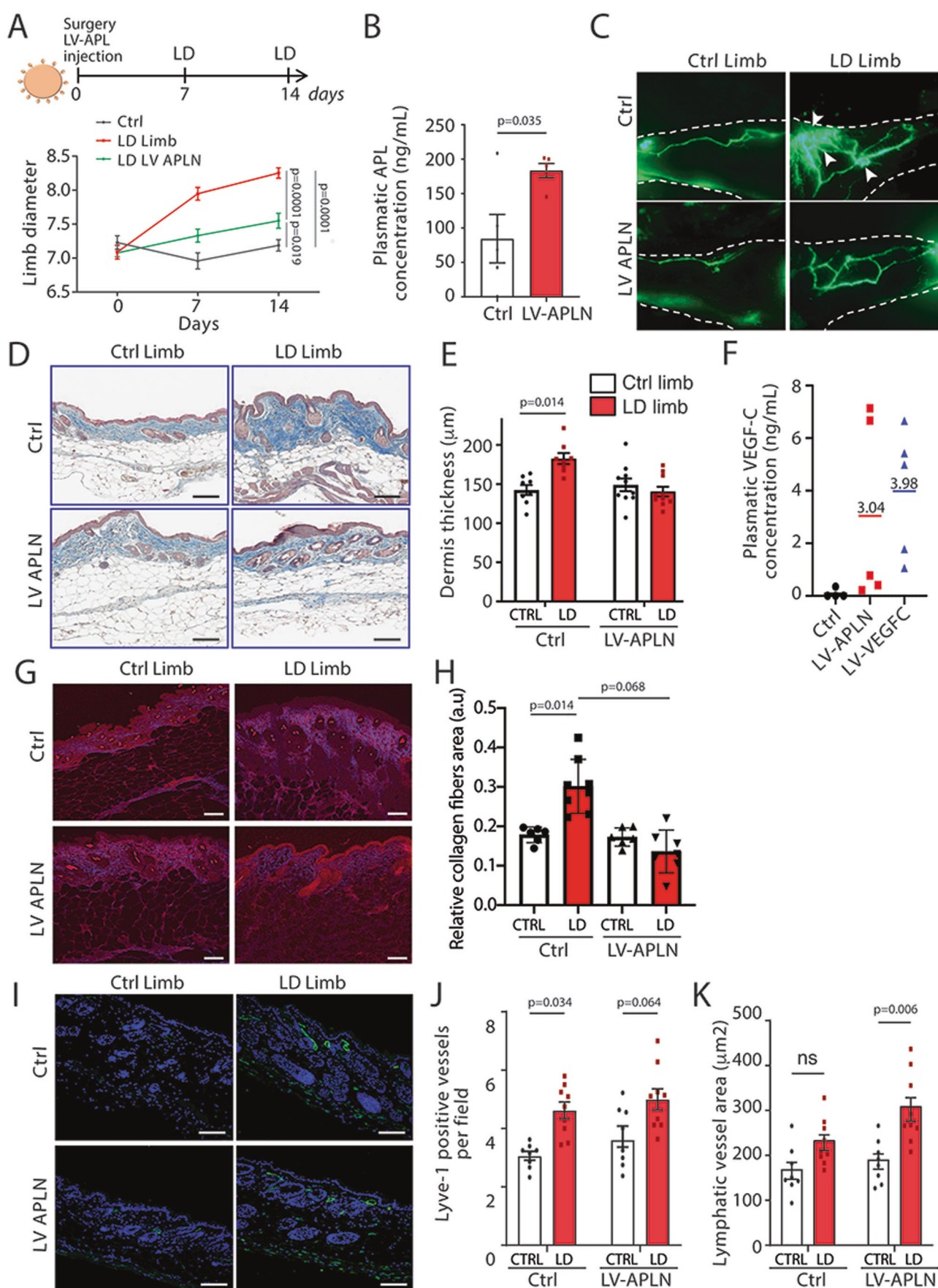

◀ **Figure 3. APLN prevents secondary LD.**

(A) Schematic of the experimental design of secondary LD model in mice injected APLN lentivector (LV-APLN). Quantification of proximal limb swelling at 7 and 14 days after surgery on the control limb, LD limb ($n = 17$) or LD treated with APLN lentivector ($n = 20$). Data represent mean ± SEM (two-way ANOVA). (B) EIA dosage of circulating APLN in plasma of control ($n = 5$) or APLN-treated mice ($n = 5$). Data represent mean ± SEM (unpaired $t$ test). (C) Lymphangiography reveals pathological remodeling of lymphatic vessels and dermal backflow in LD that is reversed by LV-APLN ($n = 10$) (scale bar: 1 mm). (D) Masson's trichrome staining of the skin from mice with LD treated or not with APLN (scale bar: 50 µm). (E) Quantification of dermis thickness. Data represent mean ± SEM ($n = 5$) (two-way ANOVA). (F). EIA dosage of circulating VEGF-C in plasma of control ($n = 5$) or APLN-treated mice ($n = 5$). (G) SHG signal from deep, collagen-rich layer within dermis (scale bar: 50 µm). (H) Quantification of the relative collagen area. Data represent mean ± SEM ($n = 6$–8) (two-way ANOVA). (I) Lyve1 immunodetection of the skin lymphangiogenesis in APLN-treated mice (scale bar: 50 µm). (J) Quantification of lymphangiogenesis in APLN-treated mice. Data represent mean ± SEM (Ctrl $n = 8$, LD $n = 9$) (two-way ANOVA). (K) Quantification of lymphatic dilatation in APLN-treated mice. Data represent mean ± SEM (Ctrl $n = 8$, LD $n = 9$) (two-way ANOVA). Source data are available online for this figure.

observed on cell junction (Fig. 5G). Importantly, eNOS phosphorylation was observed in HDLEC in response to APLN and/or VEGF-C, suggesting that both APLN and VEGF-C stimulate lymphatic dilatation through eNOS pathway (Fig. 5H,I).

## APLN stimulates lymphatic collector pumping through eNOS activation

We next explored whether APLN could control vessel dilatation, in particular regarding collecting lymphatic vessels. APLN was described to activate eNOS phosphorylation in several cellular contexts and to promote blood vessel dilatation (Dray et al, 2008; Wysocka et al, 2018). We then investigated whether APLN was able to stimulate lymphatic collecting vessel dilatation and thus lymphatic pumping (Fig. 6A; Supplemental Movies 1–3). Lymph flow is in part driven in collecting lymphatics by autonomous contraction of smooth muscle cells. To evaluate the effect of APLN on collecting vessel contractions, we used the intravital imaging method previously described (Liao et al, 2014) (Figs. 6A–C and EV4A). Number of contractions and dilatation of vessels was assessed. Interestingly, we found that APLN stimulated lymphatic pumping by increasing the collector dilatation (Fig. 6A,B) without major effect on contraction frequency (Fig. 6C). This effect was completely reversed by the L-NAME, the nitric oxide synthase (NOS) inhibitor (Figs. 6A–C and EV4A). Then, to evaluate the role of eNOS activation in response to APLN in vivo in LD context, LV-APLN-treated mice were submitted to L-NAME treatment (Fig. 6D). Limb diameter was measured to assess the edema (Fig. 6E). Interestingly, L-NAME reversed the beneficial effect of APLN on LD confirming the lymphatic pumping as a major etiology of the pathology. We also observed an increase of edema two weeks after surgery in the presence of APLN + L-NAME (Fig. 6E). Lymphangiography revealed that L-NAME treatment also reverses APLN effect on lymphatic vascular network. Indeed, in APLN + L-NAME-treated mice, we observed pathological remodeling of lymphatic vessels with dermal backflow and abnormal lymphatic branching (Fig. 6F). Capillaries were also quantified using Lyve1 immunodetection on skin sections (Fig. 6G–I). An increase of lymphangiogenesis was observed systematically in LD limb, however we did not observe any differences between conditions (Fig. 6H). Nevertheless, regarding vessel area, APLN-treated mice displayed an increased dilatation that was inhibited by L-NAME treatment restoring a similar phenotype compared to control group (Fig. 6I). We also investigated fibrosis, and surprisingly L-NAME had no effect on fibrosis (Fig. EV4B,C). Taken together, our results show that APLN prevents LD by promoting collecting vessels pumping and remodeling of lymphatic

vessels. This phenotype seems to be mediated in part by activating eNOS in lymphatic endothelial cells.

## APLN and VEGF-C exhibit a synergistic effect on the regulation of gene expression related to collecting vessel maintenance

Most of the studies aiming at regenerating the lymphatic system have focused on the VEGF-C molecule. Nevertheless, VEGF-C alone has appeared ineffective to improve lymphatic collector function in mice model of vascular injury, suggesting that it has to be combined with other molecules to fully restore the lymphatic drainage. Here we found that APLN controlled LD fibrosis, lymphatic function, and contractility of collecting vessels. Our original approach aims at combining VEGF-C with APLN to obtain a synergistic effect for the treatment of secondary LD by targeting the entire lymphatic network from capillaries to collectors. When comparing gene expression profile of APLN-, VEGF-C, or APLN + VEGF-C stimulated HDLEC, we observed similar induction of top 30 genes mostly related to extracellular matrix remodeling (Fig. 7). The majority (43) of the genes induced by VEGF-C are also induced by APLN (Fig. 7A–C) (Table 3). Half of the genes induced by the combination of APLN + VEGF-C are induced by APLN (Fig. 7D–F) (Table 4). When comparing the induction of genes shared by the two molecules, the cooperative effect of APLN and VEGF-C remained focused on genes related to the microenvironmental maintenance of the lymphatic system (collagen 1A1 and 6A) and to the maturation of VEGF-C (ADAMTS2, E2F8) (Fig. 7G). Also, 33 genes are specifically upregulated by the combination of APLN and VEGF-C (Fig. 7G). When comparing nontreated-, VEGF-C alone or APLN alone to APLN + VEGF-C combination, we found an increase of genes necessary for collecting vessel function including connexin 37 and 47 (GJA4, GJC2) and claudin 5 (CDN5), whereas angiogenic genes were downregulated (VEGFA, FLT1, KDR, KI67) (Fig. 7H,I). The expression of genes related to VEGF-C maturation (ADAMTS2, CCBE1) was improved in VEGF-C-, APLN-, and APLN-VEGF-C groups compared to WT (Fig. 7I).

## APLN-VEGF-C RNA delivery: a new therapeutic option for secondary LD

In western countries, secondary LD develops mostly after cancer treatment, which makes ethical concern for the delivery of angiogenic molecules to cancer survival patients. For security reasons, we decided to use the next generation of vector called LentiFlash® (Lf) which allows mRNA transient delivery from

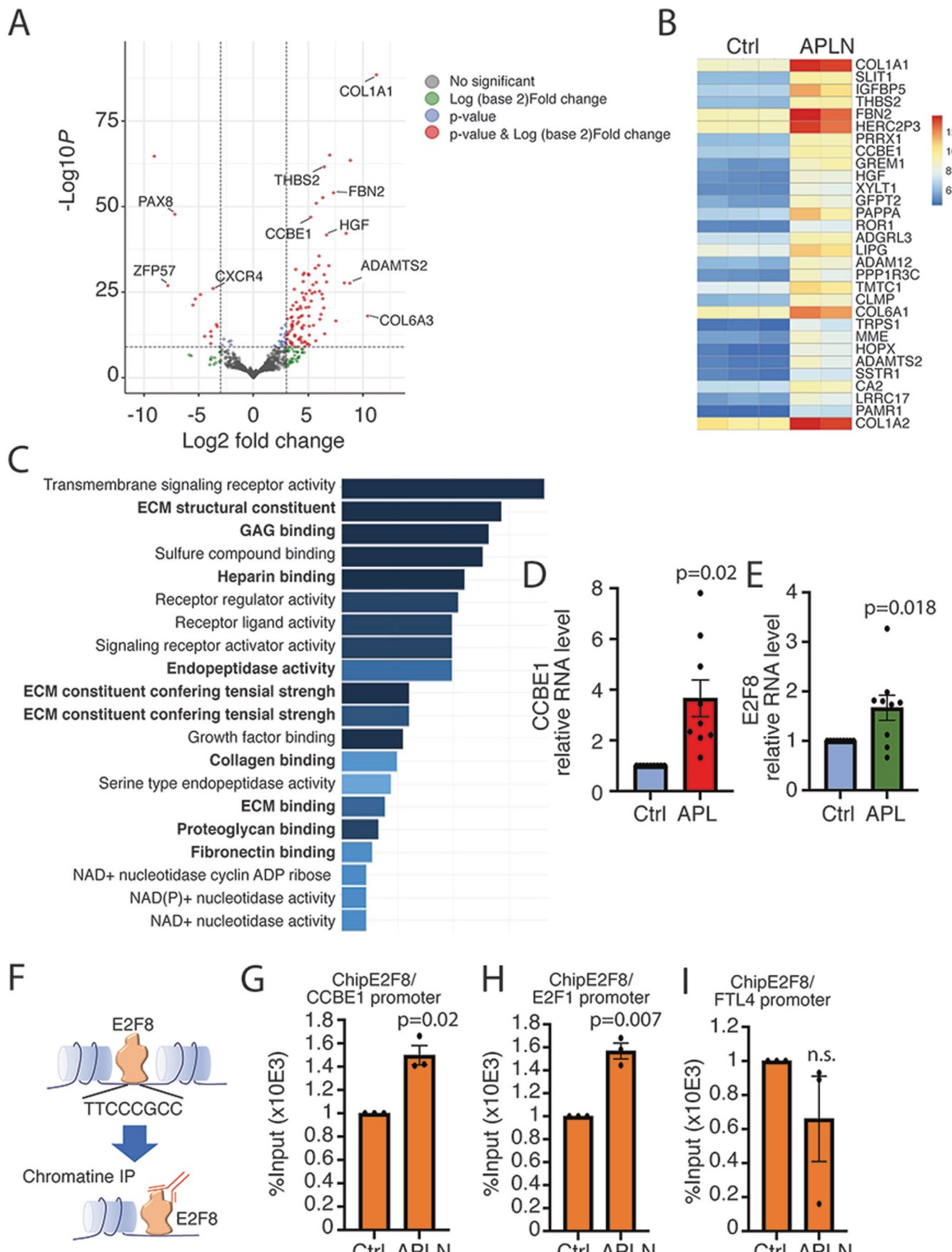

◄ **Figure 4. APLN controls lymphatic endothelial cell gene expression.**

(A) Bulk RNA sequencing in HDLEC treated with APLN-conditioned medium. Volcano plot showing log2FC (fold change) values calculated between control and APLN-treated HDLECs for 24 h. Red and blue dots: significantly (P value adjusted < 0.05) up- (log2FC > 0.5) and downregulated genes (log2FC < -0.5), respectively. (B) Heatmap of the top 30 significantly upregulated genes (P value adjusted < 0.05 and log2FC > 0.5). Expression levels are plotted as log10 normalized counts for each sample. Red represents higher FC; Dark blue represents lower FC. (C) Top significantly (FDR < 0.05) enriched Gene Ontology (GO) terms for biological processes of significantly upregulated genes after APLN treatment at 24 h time point. (D, E) qRT-PCR validation of CCBE1 (D) and E2F8 (E) in APLN-stimulated HDLEC. Data represent mean ± SEM (three independent replicates)(unpaired t test). (F) Schematic representation of chromatin immunoprecipitation (Chip) by E2F8. (G) Chip analysis of E2F8 on CCBE1 promoter. Data represent mean ± SEM (three independent replicates) (unpaired t test). (H) Chip analysis of E2F8 on E2F1 promoter. Data represent mean ± SEM (three independent replicates) (unpaired t test). (I) Chip analysis of E2F8 on Flt4 promoter. Data represent mean ± SEM (three independent replicates) (unpaired t test, ns non significant). Source data are available online for this figure.

**Table 1. Gene expression changes in response to APLN treatment.**

| Gene name | Log2 fold change | P adj |
|---|---|---|
| SELE | 2.812731725 | 0.0000000648 |
| MMP10 | 2.752225374 | 0.000369035 |
| HDAC9 | 2.741997597 | 0.004410612 |
| CRYAB | 2.635306806 | 0.000147694 |
| CLDN1 | 2.389292005 | 0.00000000477 |
| AC089983.1 | 2.36902799 | 0.000247378 |
| CCL5 | 2.276903012 | 0.000607478 |
| SERPINB2 | 2.270073695 | 0.010644328 |
| CXCL8 | 2.211382158 | 0.000798952 |
| CDCP1 | 2.201518538 | 0.000370379 |
| MYEOV | 2.18951514 | 0.007186895 |
| LTB | 2.040538173 | 0.008297478 |
| OIP5 | 1.941683429 | 0.003142472 |
| C15orf54 | 1.928461604 | 0.00857569 |
| CSF3 | 1.926259196 | 0.011845263 |
| S1PR3 | 1.904941869 | 0.0000201 |
| CPA4 | 1.89809741 | 0.000731568 |
| PLAU | 1.896943904 | 5.38E-11 |
| NRG1 | 1.896230589 | 0.000443877 |
| FGF5 | 1.885713721 | 0.000792216 |
| ICAM1 | 1.818957189 | 0.008977731 |
| CX3CL1 | 1.818549284 | 0.013746926 |
| APLN | 1.810337986 | 0.000091 |
| MAP2 | 1.801050773 | 0.000359968 |
| VCAM1 | 1.783412264 | 0.0000709 |
| IL7R | 1.776145301 | 0.0000584 |
| IL32 | 1.765738581 | 0.0000152 |
| RGS7 | 1.750200793 | 0.03119854 |
| SERPINE1 | 1.733383007 | 0.0000227 |
| AL590004.4 | 1.724948793 | 0.014089116 |
| AL121718.1 | 1.710482389 | 0.02231266 |
| KRT7 | 1.707903162 | 3.03E-47 |
| LINC01013 | 1.700691575 | 0.027792639 |
| LINC02407 | 1.69104104 | 0.015865541 |
| CD34 | 1.668560141 | 0.005604739 |
| ENOX1 | 1.668378107 | 0.00000016 |
| AC009549.1 | 1.665659241 | 0.000766355 |
| RGS4 | 1.657093433 | 7.05E-19 |
| TRNP1 | 1.633496667 | 4.68E-16 |

Bulk RNAseq analysis of the upregulated genes.

**Table 2. Gene expression changes in response to APLN treatment.**

| Gene name | Log2 fold change | P adj |
|---|---|---|
| NTN1 | −2.042642397 | 9.54E-20 |
| GPR183 | −2.040951658 | 0.015118895 |
| MATN2 | −1.912634816 | 4.06E-15 |
| ZNF853 | −1.819333215 | 0.00000158 |
| ADAMTS12 | −1.747010742 | 0.013454121 |
| ITGA8 | −1.721413706 | 0.008408822 |
| ADAMTS15 | −1.650776708 | 0.017058547 |
| STAB2 | −1.641779719 | 8.22E-10 |
| PCDH17 | −1.63376972 | 0.00000796 |
| ABCA8 | −1.588234245 | 0.011875107 |
| RASSF10 | −1.569222697 | 2.79E-11 |
| ELN | −1.548588388 | 0.002064057 |
| RASL10A | −1.510880484 | 0.003680737 |
| GRIK1 | −1.485974738 | 0.00000189 |
| AQP1 | −1.478115423 | 0.005785265 |
| FZD10-AS1 | −1.464029554 | 0.003380562 |
| STXBP6 | −1.448527106 | 0.0000383 |
| ABCG2 | −1.395464708 | 0.005472471 |
| ZDHHC8P1 | −1.372731639 | 0.044276798 |
| F8 | −1.339852658 | 0.00000000565 |
| OXTR | −1.26391024 | 0.000000446 |
| SCUBE3 | −1.258798746 | 4.39E-10 |
| SGSM1 | −1.215154068 | 0.00000000792 |
| TRMT9B | −1.197168074 | 0.002777533 |
| AC068580.1 | −1.171123379 | 0.002636392 |
| ANGPTL4 | −1.145502146 | 7.96E-09 |
| CLEC10A | −1.138370554 | 0.036713069 |
| LGR4 | −1.135924784 | 0.025769443 |
| IL33 | −1.132392151 | 0.000191547 |
| C2CD4C | −1.126040716 | 0.001399677 |
| CPNE5 | −1.119558184 | 0.000000282 |
| FZD10 | −1.119268035 | 0.000000269 |
| ALDH5A1 | −1.119096355 | 0.017283362 |
| ADD3-AS1 | −1.115145567 | 0.00000328 |
| LAMC3 | −1.108987327 | 0.00000061 |
| SIPA1L2 | −1.100969324 | 0.000366948 |
| C10orf128 | −1.098567203 | 0.019442828 |
| IGF2 | −1.092826681 | 0.003395681 |
| REEP1 | −1.08313355 | 0.0000000453 |

Bulk RNAseq analysis of the downregulated genes.

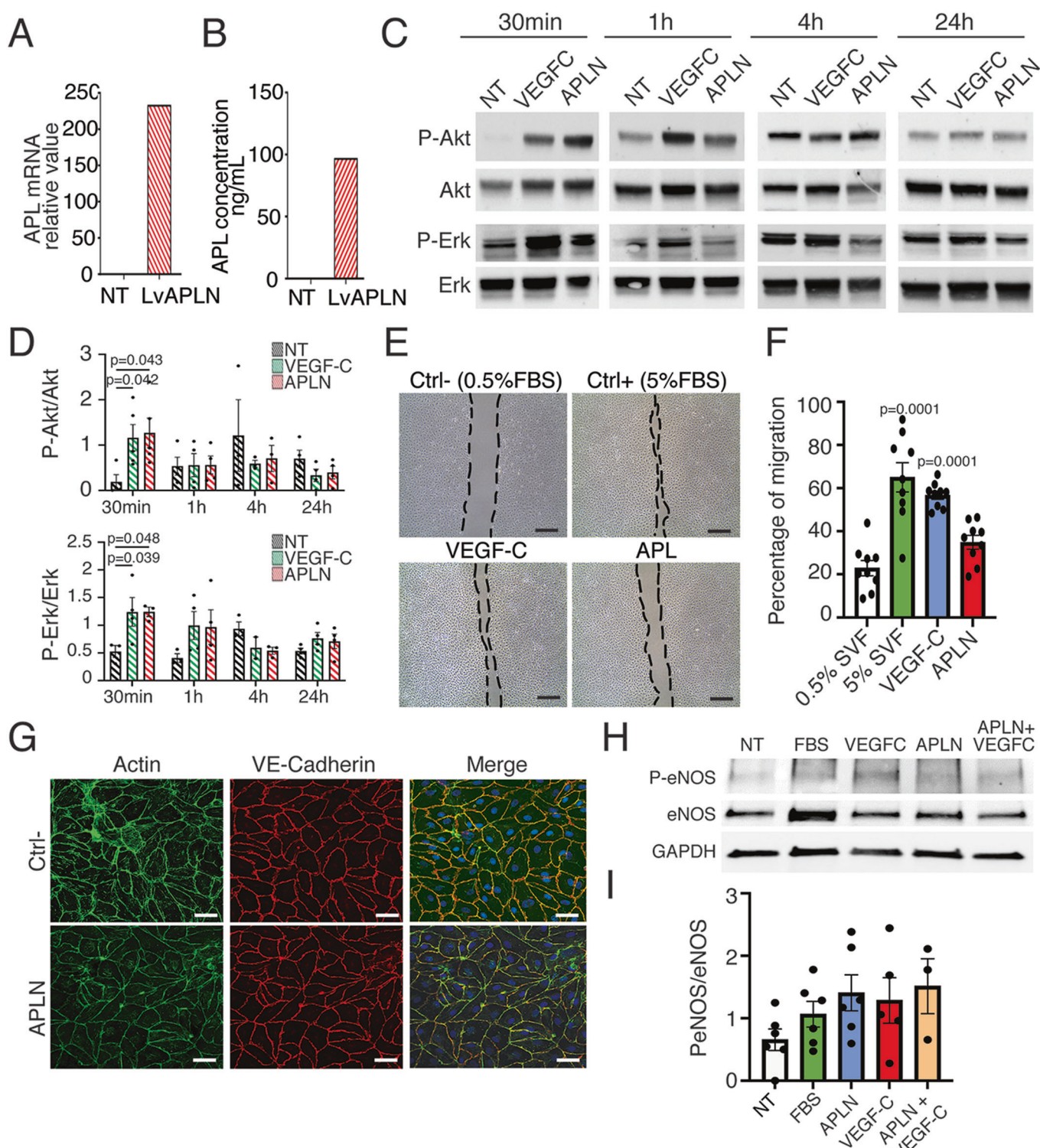

noninteegrative viral particles. Based on Lf capacity for delivering several heterologous mRNA molecules, we generated a Lf vector containing two different mRNAs coding for VEGF-C or APLN, respectively, to be injected in the mouse model of LD (Fig. 8A). Lf efficiency is highly dependent on mRNA stability compared to lentivector that induces permanent expression of the transgene without any effect on immune cell populations of platelet numbers

(Fig. EV5A,B). We therefore first confirmed the presence of circulating APLN (Fig. 8B) and VEGF-C (Fig. 8C) by ELISA, measurable 48 h after injection. We only observed a partial inhibition of limb swelling using APLN or VEGF-C mRNA alone (Fig. 8D,E). This could be expected, due to the restricted time of molecule expression. However, the APLN-VEGF-C double mRNA Lf completely abolished limb swelling (Fig. 8F,G), reduced dermal

◄ **Figure 5.   APLN plays a role in LD through Akt/eNOS activation in lymphatic endothelial cells.**

HDLEC were treated in vitro with conditioned media of NIH3T3 cells infected APLN lentivector. (**A**) Relative expression of APLN in NIH3T3 evaluated by RT-qPCR on NIH3T3 transduced by APLN lentivector. Data represent pool of Lv-transduced cells. (**B**) Expression of APLN in conditioned media evaluated by EIA. Data represent pool of Lv-transduced cells. (**C**) Representative phospho-AKT/AKT and phosphor-Erk/Erk immunoblots of HDLEC treated with FBS, conditioned medium containing VEGF-C or APLN. (**D**) Graphs represent quantification of phospho/total protein ratio of at least three independent experiments. All graphical data are mean ± SEM. *$P < 0.05$, two-way ANOVA. (**E**) Representative images of scratch wound healing assay on HDLEC stimulated by VEGF-C or APLN. (scale bar: 100 μm). (**F**) Quantification of migration. Data represent mean ± SEM ($n = 9$) (one-way ANOVA). (**G**) F-actin and VE-Cadherin immunostaining of HDLEC after APLN treatment reveals no effect on lymphatic endothelial monolayer junctions ($n = 3$) (scale bar: 25 μm). (**H**) Representatives phospho-eNOS/eNOS immunoblots of HDLEC treated with FBS, conditioned medium containing VEGF-C or APLN. (**I**) Graphs represent quantification of phospho/total protein ratio of at least three independent experiments. Data represent mean ± SEM ($n = 6$) (one-way ANOVA). Source data are available online for this figure.

backflow (Fig. 8H) and restored the lymphatic perfusion in the lymphedematous limb (Fig. 8I). This was associated with an increase in lymphatic vessel diameter (Fig. 8I). Finally, to investigate whether Apelin-VEGF-C mRNA could be a curative treatment for LD, we injected mice which developed LD (10 days after surgery) (Fig. 8J). In that context, Lf vector reversed LD swelling to go back to normal after 11 days. These data showed the synergistic effect of Apelin and VEGF-C and demonstrated that this combination generates a significant therapeutic benefit despite the transient expression of the two transgenes, providing a perspective of LD treatment using nonintegrative RNA delivery vectors for patients who develop LD after cancer treatment.

## Discussion

Despite large advances in the past decades for the understanding of the molecular mechanisms that drive the lymphatic function, LD, the most predominant pathology associated with lymphatic dysfunction remains an unmet medical need (Mercier et al, 2019). It is a chronic condition that affects millions of people worldwide. Many factors contribute to the etiology of the disease. Primary LD, an inherited disease, is induced by genetic mutation, whereas secondary LD occurs after cancer treatment or filarial infection (Mortimer and Rockson, 2014; Rockson, 2018). However, they all lead to comparable clinical signs: an accumulation of fluid and fat in the limb associated with fibrosis and hypervascularized dermis characterized by tortuous and leaky capillaries and hypoperfusion of deeper collecting vessels. Lymphoscintigraphies show a severe reduction of lymph node perfusion demonstrating that the lymphatic collecting vessels are still present, but cannot collect and drive the lymph properly. These observations support the regenerative therapeutic strategy aiming at combining molecules to 1/normalize the capillary territories and 2/regenerate the lymphatic pumping in deeper adipose depots. Whereas it is now well established that VEGF-C, the major lymphangiogenic growth factor, is the best candidate the restore the lymphatic capillary network (Hartiala et al, 2020a), its role on collecting vessels remains less effective. Collecting vessels develop in an integrated adipose environment that is considerably modified during LD. In particular, adipose tissue synthesize many adipokines involved in the blood and lymphatic vessel integrity. It is therefore tempting to speculate that changes in adipokine production may affect the lymphatic collecting function. Among them, APLN has been described to be a key factor for stimulating LEC function (Kim et al, 2014). APLN is a bioactive peptide that induces signaling after binding to its G-protein-coupled receptor APJ located at the surface of LEC. It stimulates lymphangiogenesis in cancer and participates to the

restoration of precollecting lymphatics shape after myocardial infarction (Tatin et al, 2017). In addition to its effect on the endothelial monolayer, APLN is a robust antifibrotic molecule (Huang et al, 2016; Renaud-Gabardos et al, 2018). Importantly, it was recently found to promote the non-sprouting expansion of vessels in the intestinal crypt (Bernier-Latmani et al, 2022).

By performing gene expression analysis of dermolipectomies from women who developed secondary LD after breast cancer, we identified a significant decrease in APLN expression in LD. The crucial role of APLN in LD was confirmed in APLN-KO mice that exhibit an aggravation of LD that can be rescued by an APLN-expressing lentivector. In the mouse model of LD, we identified that APLN improves LD condition by acting on two major hallmarks of the pathology: lymphatic function and tissue fibrosis. In LEC, APLN controls the expression of genes involved in extracellular matrix remodeling in line with its effect on tissue fibrosis. Interestingly, APLN also strongly stimulated the expression of CCBE1, a protein involved in the proteolytic activation of VEGF-C by ADAMTS3 (Jha et al, 2017). This could in part explain the increase of circulating VEGF-C concentration observed after APLN treatment. Importantly, in human, mutations in *CCBE1* were found to cause Hennekam syndrome, a congenital disease leading to LD, lymphangiectasia, and heart defects (Alders et al, 2013). Mechanistically, we found that CCBE1 gene expression is controlled by APLN that directly increase the fixation of E2F8, transcription factor on its promoter. Altogether, these data reinforce the APLN as a key factor in restoring the lymphatic function in LD.

Also, we found that the effect of APLN on the lymphatic collecting pumping was directly controlled by eNOS. NO production participates in the endothelial homeostasis by controlling the modulation of vascular tone as an adaptation of flow (Dimmeler et al, 1999). The endothelial NOS (eNOS) also regulates lymphatic homeostasis. In a mouse model of fibrosarcoma, eNOS mediates VEGF-C-induced lymphangiogenesis and tumor lymphatic metastasis (Lahdenranta et al, 2009). Other studies have shown that eNOS affects the lymph flow via the collecting lymphatics, without affecting the diameter of capillaries (Hagendoorn et al, 2004). Also, NO bioavailability in pulmonary lymphatics was found to be impaired in limbs that exhibit chronically increased pulmonary blood and lymph flow (Datar et al, 2016). Here, we identified that the effect of APLN on the lymphatic collectors pumping is mediated by eNOS. APLN was previously described to modulate the aortic vascular tone by increasing the phosphorylation of Akt and eNOS in diabetic mice (Zhong et al, 2007). We found that the beneficial effect of APLN on lymphatic collecting vessels is mediated by this pathway, suggesting that APLN can be at the origin of NO-mediated lymphatic pumping in many organs. The Akt-phosphorylation was found in a lesser extent than the phosphorylation induced by VEGF-C; however, it seems to be efficient to mediate its

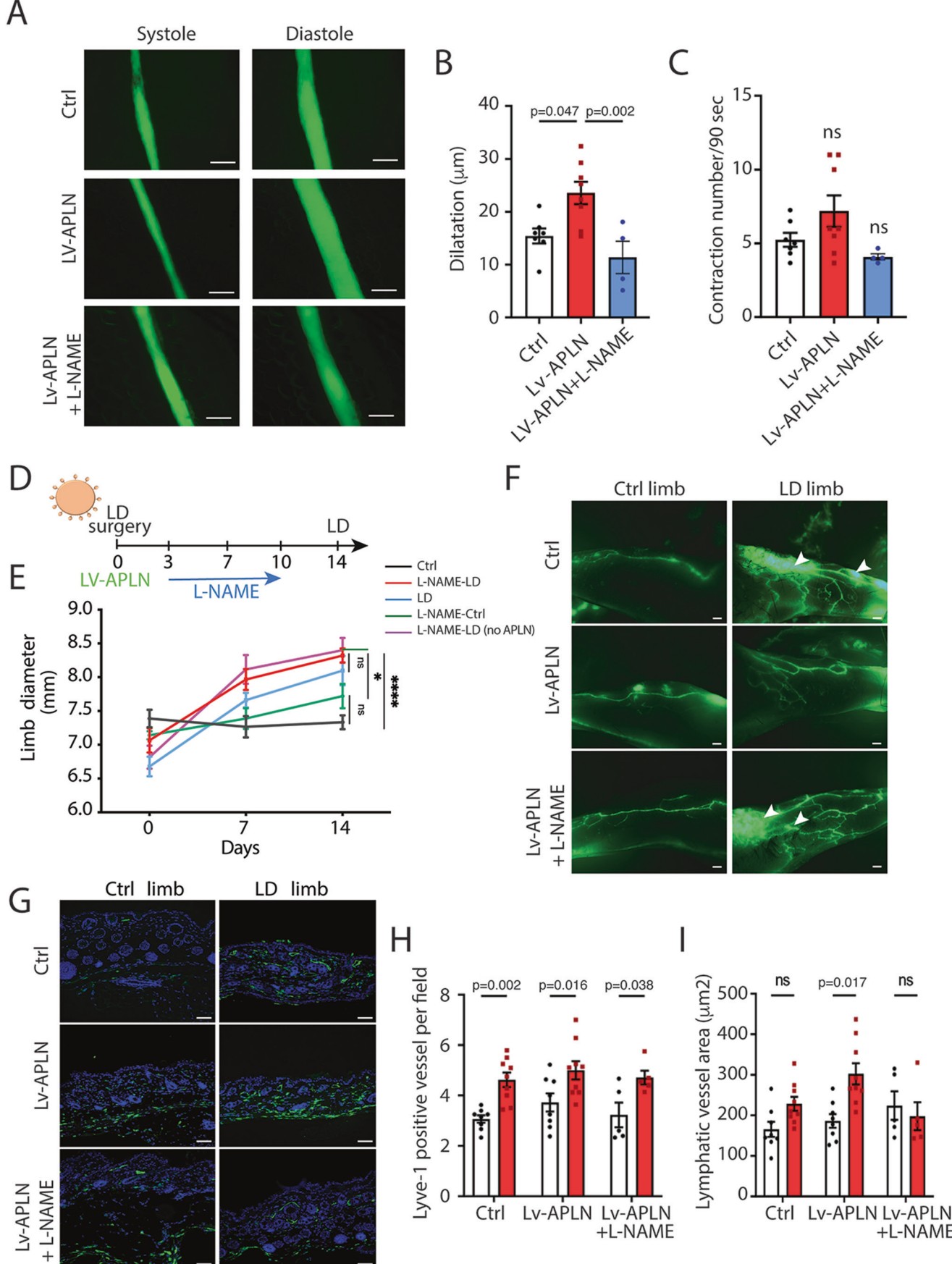

**Figure 6. APLN-induced vasodilatation of lymphatic vessels is mediated by eNOS signaling.**

(A) Contractile activity of collecting lymphatic vessels in control mice ($n = 7$) treated with APLN lentivector ($n = 8$) and L-NAME ($n = 4$) was investigated by filming autonomous collecting vessel contraction in vivo (scale bar: 100 μm). (B, C) graphs represent the number of vessel contractions per film (90 s) and the dilatation of collecting lymphatic vessels (differences between maximum and minimum diameter). Data represent mean ± SEM ($n = 8$) (one-way ANOVA). (D) Schematic of the experimental design of secondary LD mice model. (E) Quantification of proximal limb swelling at 7 and 14 days after surgery on control limb, LD limb ($n = 9$) or LD treated with APLN lentivector ($n = 10$) followed or not by treatment with L-NAME ($n = 5$). Data represent mean ± SEM (two-way ANOVA). (F) Representative images of lymphangiography from mice treated with LV-APLN and L-NAME. (scale bar: 1 mm). (G) Skin sections were stained with Lyve1 (green) to assess the number of lymphatic capillaries in control ($n = 9$), LV-APLN ($n = 9$) or LV-APLN + L-NAME ($n = 5$) treated mice. (scale bar: 50 μm). (H, I) Graphs show the number of lymphatic vessels (Lyve1[+]) (H) and the vasodilatation (vessel area) (I) according the experimental conditions. Data represent mean ± SEM ($n = 8$) (two-way ANOVA). Source data are available online for this figure.

biological effects and suggests that APLN and VEGF-C have complementary effect to mediate biological actions. Therefore, we propose to evaluate the effect of APLN coupled to VEGF-C for the treatment of LD. However, an important ethical issue in treated cancer survivor patients is to reactivate the tumor with pro-lymphangiogenic therapy, even with more than five years without any recurrence. Therefore, the use of long-term delivery tools such as integrative lentiviral or AAV vectors rapidly appeared as sub-optimal solution for treatment delivery due to uncontrolled protein expression duration and insertional mutagenesis risk. Another issue is the brief plasma APLN half-life which is less than 5 min (Japp and Newby, 2016). This could have been compensated by successive injections in the limb, however it would significantly improve the risk of infections and desmoplastic reaction, which are often seen in LD patients. We then decided to use a biological RNA delivery approach able to limit the dose requirement of a synthetic RNA-based therapeutic drug. LentiFlash® Technology, based on a novel class of chimeric lentiviral platform, allows the delivery of transient multiple biological mRNA molecules (Prel et al, 2015). LentiFlash® is constructed using a bacteriophage coat protein and its cognate 19-nt stem loop, to replace the natural lentiviral Psi-mediated packaging system, in order to achieve active biological mRNA packaging into the lentiviral particles. Associated to a total removing of HIV sequences on the RNA packaged into lentiviral particle (LTR, PBS, RRE sequences), it enables the encapsidation of multiple and heterologous RNA sequences into the same LentifFlash particle without any risk of integration into the host genome. The present study shows that single APLN mRNA or VEGF-C mRNA alone delivery mediated by LentiFlash® exhibits less efficiency in reducing LD compared to integrative lentivector. As shown in Fig. EV5, VEGF-C mRNA delivery alone or APLN mRNA delivery alone injected at the time of surgery did not reduce LD in three independent experiments. In accordance with the 3R (Replacement Refinement Reduction) rules from EU ethic committee, we decided not to pursue experiments with single mRNA delivery. In addition, phase I and II clinical trials have previously examined the effectiveness of Lymfactin®, an experimental adenoviral-based gene therapy vector that encodes human VEGF-C, for the treatment of breast cancer-related secondary LD (Hartiala et al, 2020b). No adverse events were recorded at the 24-month follow-up; however, the drug's development was stopped since the phase II double-blind, randomized, placebo-controlled, multicenter clinical trial yielded inconclusive results (Leppapuska et al, 2022).

However, when combined to VEGF-C, double mRNA delivery completely abolished LD and restored the lymphatic flow in the limb showing that mRNA delivery strategy allows enough synthesis of the two proteins to observe a beneficial effect.

In the past 2 years, we have seen the emergence of a novel class of mRNA vaccine, a highly efficient and low-toxic vector. We believe that mRNA can treat many diseases including LD, a

pathology with currently no treatments, in a different way than traditional medicine. The plasticity of LentiFlash® allows the transient delivery of two different mRNA molecules, allowing here to stimulate the synergistic effect of APJ, a G-protein-coupled receptor with VEGFR3, a tyrosine kinase receptor. Based on the fact that LD remains a multifactorial pathology with lymphatic endothelial dysfunction, AT accumulation, and fibrosis, we are convinced that multiple therapy will be the solution to cure this harmful condition. Therefore, we proposed to use the APLN-VEGF-C LentiFlash® vector for a phase I/II gene therapy clinical trial called Theralymph that is in the process of being launched in Toulouse University Hospital where our laboratory is located.

## Methods

### Human tissue specimens

Samples were obtained from archival paraffin blocks of 16 lipodermectomy specimens, obtained from patients with secondary LD, treated at Toulouse University Hospital, France, between 2015 and 2016.

Patients with a history of 1 to 12 years LD presented stage 2 LD according to the classification of the International Society of Lymphology (ISL). Eligible patients had a history of unilateral non-metastatic breast cancer without recurrence for more than 5 years. The main clinical parameters used to diagnose a significant LD included a clinical upper limb edema with a volume difference between the upper limb affected and the other upper limb of 10% or 200 milliliters (mL).

Samples were selected as coded specimens under a protocol approved by the INSERM Institutional Review Board (DC-2008-452), the Research State Department (Ministère de la recherche, ARS, CPP2, authorization AC-2008-452) and the Ethic Committee. Informed consent was obtained from all subjects. Experiments are conformed to the principles set out in the WMA Declaration of Helsinki and the Department of Health and Human Services Belmont Report.

When available, some control arm tissue samples removed for esthetical purpose in the same patients, were studied.

### Lymphofluoroscopy

Near-infrared fluorescence lymphatic imaging is used to visualize the initial and conducting lymphatics. In total, 100 μg of indocyanine green (Pulsion®) was diluted in a volume of 0.5 mL of pure water before intradermal injection into the first interdigital space. Fluorescence imaging of the lymphatic flow was observed by fixing the camera (Photo Dynamic Eye, Hamamatsu®) 15 cm above

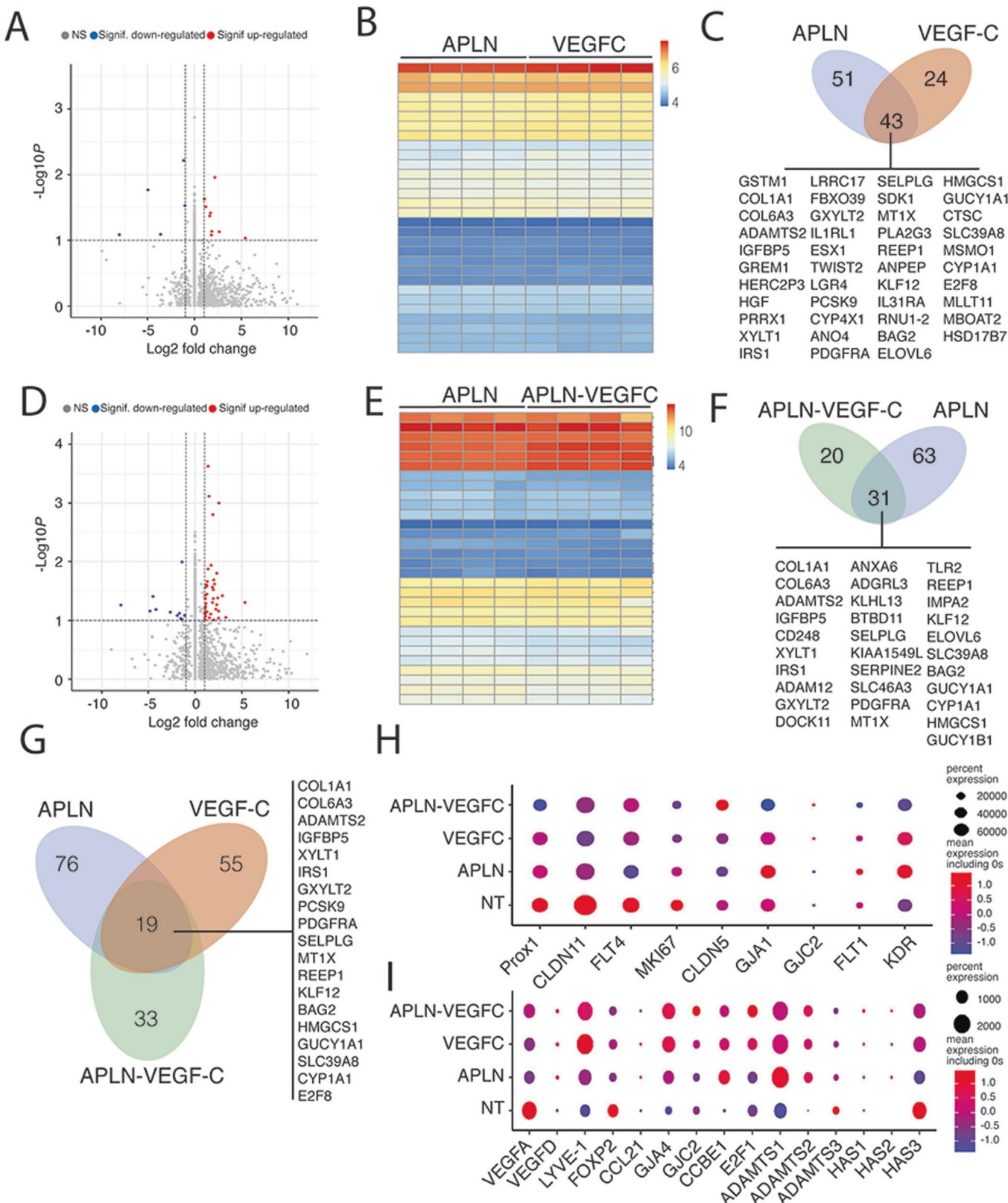

**Figure 7.  APLN and VEGF-C exhibit complementary effects on lymphatic endothelial cells.**

Comparison of bulk RNA sequencing in HDLEC treated with APLN-, VEGF-C-, or APLN + VEGF-C-conditioned media. (**A–C**) Bulk RNA sequencing in HDLEC treated with APLN- and VEGF-C-conditioned medium. (**A**) Volcano plot showing log2FC (fold change) values calculated between APLN- and VEGF-C-treated HDLECs. Red and blue dots: significantly ($P$ value adjusted $< 0.05$) up- (log2FC $> 0.5$) and downregulated genes (log2FC $< -0.5$), respectively. (**B**) Heatmap comparison of the top 30 significantly regulated genes in APLN- and VEGF-C-treated HDLEC. (**C**) Schematic representation of the number of genes upregulated (43) by both APLN and VEGF-C. (**D, E**) Bulk RNA sequencing in HDLEC treated with APLN and APLN-VEGF-C-conditioned medium. (**D**) Volcano plot showing log2FC (fold change) values calculated between APLN- and APLN-VEGF-C-treated HDLECs. Red and blue dots: significantly ($P$ value adjusted $< 0.05$) up- (log2FC $> 0.5$) and downregulated genes (log2FC $< -0.5$), respectively. (**E**) Heatmap comparison of the top 30 significantly regulated genes in APLN- and APLN-VEGF-C-treated HDLEC. (**F**) Schematic representation of the number of genes upregulated (31) by both APLN and APLN-VEGF-C. (**G**) Schematic representation of the number of genes upregulated (19) by both APLN, VEGF-C, and APLN + VEGF-C. (**H**) Dot plots showing the expression of known lymphatic markers in nontreated (NT), APLN, VEGF-C and APLN + VEGF-C-treated HDLEC. (**I**) Dot plots showing the expression of known lymphatic markers in NT, APLN, VEGF-C, and APLN + VEGF-C-treated HDLEC. Source data are available online for this figure.

the investigation field. Indocyanine green lymphography findings are classifiable into two patterns: normal linear pattern and abnormal dermal backflow pattern.

## Lymphoscintigraphy

Lymphoscintigraphy was used to diagnose the severity of LD. It is a low-radiation examination, forbidden during pregnancy and breastfeeding. Bilateral hypodermal injections were administered between the first and second fingers. The large size of 99m-technetium radiolabeled nanocolloidal albumin were selectively entrapped by lymphatic capillaries and then drained by the lymphatic system. It enabled comparative, functional and bilateral evaluation of the two upper limb including axillary lymph node uptake, lymphostasis, dermal backflow, and rerouting into the deep lymphatic system into the epitrochlear lymph node.

## Mouse model of LD

Mouse procedures were performed in accordance with EU and national regulation. C57Bl/6 mice were provided by Envigo. Mice were held under specific pathogen-free conditions in the animal facility of the Inserm Crefre US06 on a 12:12 light: dark cycle. In all cases, experimental and control animals were of the same age and gender. All experiments have been approved by the local branch Inserm Rangueil-Purpan of the Midi-Pyrénées ethics committee. Secondary LD was established as previously described (Morfoisse et al, 2018). Briefly, LD was established in the left upper limb of 6-week-old C57Bl/6 female mice. A partial mastectomy of the second mammary gland was performed in association with axillary and brachial lymphadenectomy. Limb size was measured over time in the axillary region using caliper. Mice sustained edema for a period of 2–4 weeks. The day of the surgery, vehicle or APLN lentiviral vector was injected intradermally in the LD limb (three injections of 2 μL 10⁶TU/mL). For LentiFlash® vector, vehicle, APLN, VEGF-C, and APLN-VEGF-C Lf vectors were injected intradermally into the LD limb (200 ng of p24 splitted into three injections of 2 μL each). For L-NAME (N$^G$-nitro-L-arginine methyl ester) (Sigma) treatment, L-NAME was resuspended in water (1 mg/ml) and mice were allowed to drink freely for 7 days. All mouse trials were conducted with at least four mice per group. Mice were randomized between treatment allocations. No blinding was performed in our experiments.

## Lymphangiography

Two weeks after surgery, mice were anesthetized with an intraperitoneal injection of ketamine (100 mg/kg) (Zoletil 100,

Virbac) and xylazine (10 mg/kg) (Rompun 2%, Bayer). FITC-Dextran (70 kDa, 2 mg/mL, Sigma) was injected into the footpad of LD and control limb. The fluorescent molecule was taken up by the lymphatics and excluded from the blood vessels. After 5 min, the skin was analyzed under the modular stereo microscope discovery V12 stereo (Zeiss).

## Histology

Skin of LD and control tissues from human and mice limb were embedded in paraffin and sectioned on a microtome. In all, 5-μm sections were cut and placed on Superfrost Plus slides. Tissue was deparaffinized, rehydrated, and antigen unmasking was realized with pH 9 Tris solution (H-3301, Vector Laboratories) 5 min three times in a microwave. After cooling, slides were washed in PBS and then blocked with 5% BSA solution at room temperature in a humid chamber. Sections were incubated with primary antibodies O/N at 4 °C (Rabbit anti-human Lyve1, Fitzgerald (1/200); Goat anti-murine Lyve1, R&D AF2125 (1/100); rabbit anti-CD31, abcam Ab28364 (1/50)) and washed three times in PBS. Sections were incubated with the corresponding secondary antibodies conjugated to Alexa -488 or -594 for 1 h at room temperature at 1/400 dilution. For fluorescence, nuclei were stained with DAPI. Images were acquired using inverted microscope (Leica, DMi8). Images were analyzed with Fiji software.

## Evaluation of fibrosis

Dermis fibrosis was evaluated with Masson's trichrome coloration (MST-100T, Cliniscience). Skin flap sections were deparaffinized and tissue were stained according to the manufacturer's recommendations. Images were acquired on a nanozoomer slide scanner. Dermis size quantification was performed using at least 20 measurements of the length between epidermis and hypodermis per field.

## SHG imaging

Skin of LD and control tissues from mice limb were embedded in paraffin and sectioned on a microtome. In all, 30-μm sections were cut and placed on Superfrost Plus slides. Tissue were deparaffinized as described in the "Histology" part and used for SHG analysis. Acquisitions were performed using a Bruker (Billerica, Massachusetts, United States) 2P Plus two-photon microscope. The microscope was equipped with a Coherent (Santa Clara, California, United States) Chameleon Discovery laser, and an Olympus (Shinjuku, Tokyo, Japan) 20× NA:1 objective. We utilized a laser wavelength of 900 nm and collected the Second Harmonic

**Table 3. Gene expression changes in response to VEGF-C treatment.**

| Gene name | Log2 fold change | P adj |
|---|---|---|
| COL3A1 | 13.51297701 | 0.000494047 |
| COL1A1 | 11.04361238 | 0.000415896 |
| COL6A3 | 9.413802034 | 0.00470325 |
| ADAMTS2 | 8.27465003 | 0.034461364 |
| IGFBP5 | 7.452186173 | 0.008385024 |
| CD248 | 5.893908153 | 0.033065433 |
| TMEM200A | 5.861334143 | 0.009638981 |
| NTM | 5.5709033 | 0.023355587 |
| CARMN | 4.915481566 | 0.008503535 |
| XYLT1 | 4.510397241 | 0.00054922 |
| IRS1 | 4.395069034 | 0.007227541 |
| GPAT2P1 | 4.079233898 | 0.047483092 |
| ADAM12 | 3.908667028 | 0.000949715 |
| GXYLT2 | 3.329572904 | 0.046590834 |
| DOCK11 | 3.144087614 | 0.039312035 |
| ANXA6 | 3.026947231 | 0.007227541 |
| ADGRL3 | 2.897773947 | 0.023355587 |
| PCSK9 | 2.804968682 | 0.002364173 |
| KLHL13 | 2.75082358 | 0.036755614 |
| HIST1H3J | 2.548542987 | 0.0000135 |
| HIST1H1A | 2.494982926 | 0.00000771 |
| BTBD11 | 2.419793652 | 0.039530643 |
| SELPLG | 2.40537126 | 0.009804674 |
| KIAA1549L | 2.391485078 | 0.039186999 |
| SERPINE2 | 2.239191837 | 0.049204756 |
| SLC46A3 | 2.22965513 | 0.032557454 |
| HIST2H2BF | 2.069223465 | 0.034938273 |
| HIST2H3D | 2.053137471 | 0.010150563 |
| PDGFRA | 2.022277061 | 0.041201923 |
| HIST1H2BL | 1.995288276 | 0.032766325 |
| SSUH2 | 1.90226196 | 0.015561785 |
| MT1X | 1.796160767 | 0.000834822 |
| TLR2 | 1.713696714 | 0.032557454 |
| REEP1 | 1.702904635 | 0.000787739 |
| HIST1H3D | 1.645591486 | 0.000891381 |
| IMPA2 | 1.623196651 | 0.038381511 |
| NMRAL2P | 1.547984923 | 0.004786025 |
| KLF12 | 1.527556133 | 0.047483092 |
| ELOVL6 | 1.464493319 | 0.002051263 |

Bulk RNAseq analysis of the upregulated genes.

**Table 4. Gene expression changes in response to APLN-VEGF-C treatment.**

| Gene name | Log2 fold change | P value |
|---|---|---|
| GSTM1 | 24.6440679 | 2.09E-11 |
| COL3A1 | 12.5362136 | 0.00000284 |
| COL1A1 | 10.1028955 | 0.000003 |
| COL6A3 | 9.35723896 | 0.0000122 |
| ADAMTS2 | 7.82327731 | 0.00017456 |
| IGFBP5 | 7.65579404 | 0.0000501 |
| GREM1 | 7.34519332 | 0.0000468 |
| AF165147.1 | 5.94252292 | 0.000026 |
| NTM | 5.8860802 | 0.00003 |
| HERC2P3 | 5.82767126 | 0.0003209 |
| HGF | 5.66483328 | 0.00017597 |
| PRRX1 | 4.7327915 | 0.00015021 |
| XYLT1 | 4.59041849 | 0.00021529 |
| IRS1 | 4.19232602 | 0.0000112 |
| LRRC17 | 4.10894287 | 0.00015412 |
| GPAT2P1 | 3.97643673 | 0.0000696 |
| FBXO39 | 3.75275673 | 0.00024014 |
| GXYLT2 | 3.59872809 | 0.00012651 |
| IL1RL1 | 3.23930623 | 0.00033524 |
| ESX1 | 2.8650003 | 0.00017638 |
| TWIST2 | 2.74510441 | 0.00015814 |
| LGR4 | 2.58878342 | 0.00016059 |
| PCSK9 | 2.58105772 | 0.000000742 |
| CYP4X1 | 2.50782533 | 0.0002746 |
| ANO4 | 2.49821267 | 0.00024518 |
| HIST1H1A | 2.34992625 | 9.54E11 |
| PDGFRA | 2.33585253 | 0.00010922 |
| HIST1H3J | 2.33141774 | 9.00E-12 |
| SELPLG | 2.24774514 | 0.0000366 |
| SDK1 | 2.23247851 | 0.00033843 |
| HIST2H2BF | 2.11705463 | 0.000000176 |
| MT1X | 2.00950211 | 5.24E-10 |
| PLA2G3 | 2.00032749 | 0.0000282 |
| HIST1H2BL | 1.96650297 | 0.000000719 |
| REEP1 | 1.80691562 | 0.00000152 |
| HIST2H3D | 1.7979865 | 0.00016134 |
| ANPEP | 1.75818463 | 0.00031543 |
| KLF12 | 1.69231642 | 0.0000755 |
| AK7 | 1.63217642 | 0.00016331 |
| IL31RA | 1.60992486 | 0.0000279 |
| RNU1-2 | 1.47684928 | 0.0000161 |
| BAG2 | 1.47215967 | 0.0000597 |

Bulk RNAseq analysis of the upregulated genes.

Generation (SHG) emission at 450 nm. Z stacks were acquired with a step size of 1 μm. Collagen fibers quantification was performed using at least five measurements per skin section. This analysis was performed on at least six different mice per condition.

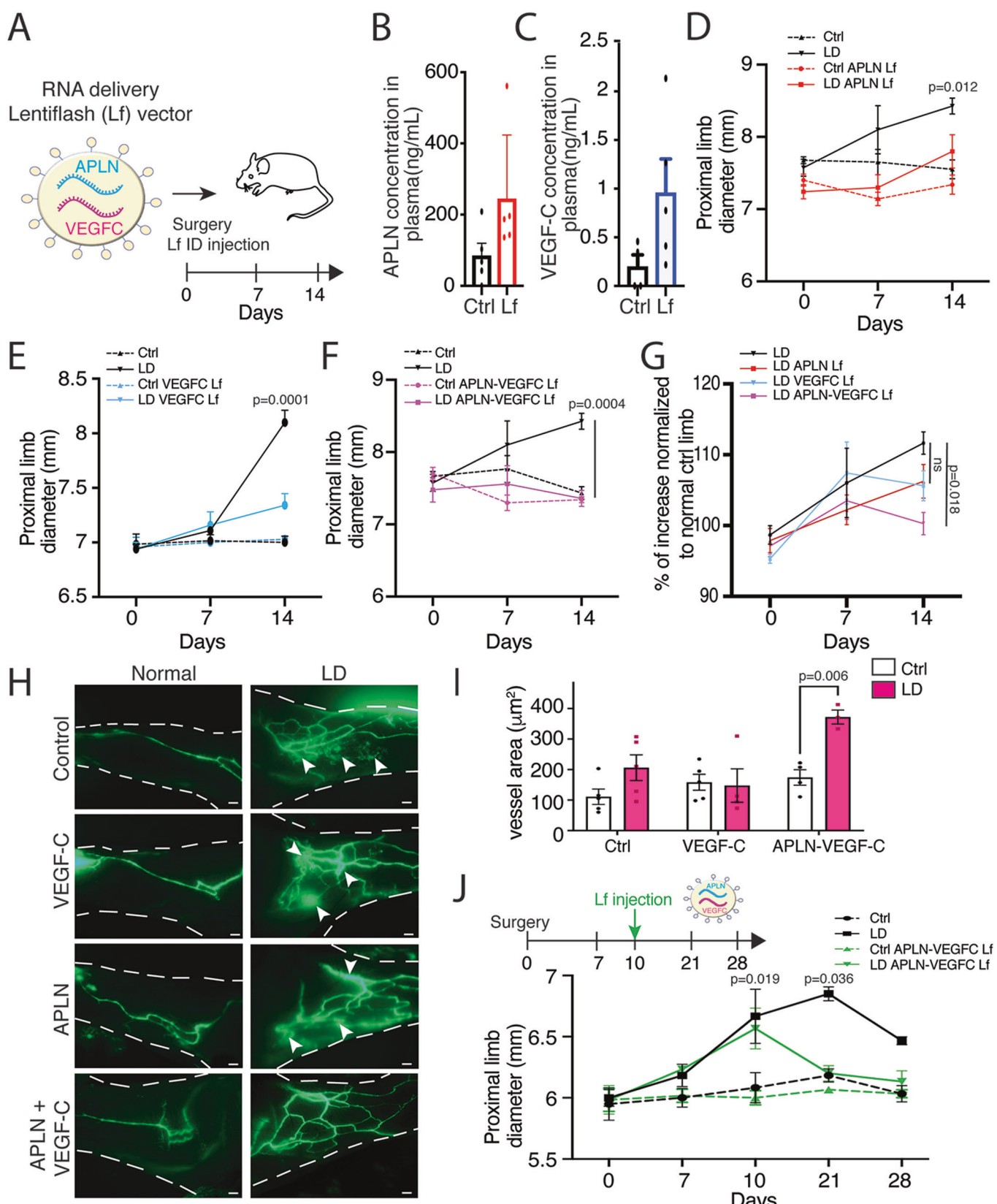

**Figure 8. APLN-VEGF-C mRNA delivery: a new treatment option for LD.**

(A) Schematic representation of the experimental procedure. Lentiflash-containing 2 mRNA (APLN and VEGFC) is injected intradermally at the time of the surgery. (B, C) EIA dosage of circulating APLN (B) and VEGF-C (C) in plasma of control ($n = 5$) or LentiFlash®-treated mice Data represent mean ± SEM (5 independent replicates) (unpaired $t$ test). (D) Quantification of proximal limb swelling at 7 and 14 days after surgery on control and LD limb from nontreated mice (black) or mice treated with APLN LentiFlash® vector (red). Data represent mean ± SEM ($n = 10$) (two-way ANOVA). (E) Quantification of proximal limb swelling 7 and 14 days after surgery on and LD limb from nontreated mice (black) or mice treated with VEGF-C LentiFlash® vector (blue). Data represent mean ± SEM ($n = 10$) (two-way ANOVA). (F) Quantification of proximal limb swelling 7 and 14 days after surgery on control and LD limb from nontreated mice (black) or mice treated with APLN-VEGF-C LentiFlash® vector (pink). Data represent mean ± SEM ($n = 10$) (two-way ANOVA). The same control group was used in 8D and 8 F. (G) Percentage of increase in LD limb compared to the normal limb on the same mouse after treatment with APLN, VEGF-C, or APLN + VEGF-C LentiFlash® vector. Data represent mean ± SEM ($n = 10$) (two-way ANOVA). (H) Representatives images of lymphangiography from mice treated with VEGF-C-, APLN-, or APLN-VEGF-C LentiFlash® vectors. (scale bar: 1 mm). (I) Quantification of lymphatic dilatation in APLN-VEGF-C-treated mice. Data represent mean ± SEM ($n = 9$) (two-way ANOVA). (J) Quantification of proximal limb swelling in mice treated with APLN-VEGF-C LentiFlash® vector after LD development (10 days post surgery). Data represent mean ± SEM ($n = 8$) (two-way ANOVA). Source data are available online for this figure.

## Collecting vessel contraction measures

Vessel contraction measurement of afferent collecting lymphatic vessels to the popliteal lymph node (PLN) was performed as described previously (Liao S et al, 2014). Briefly mice were anesthetized with an intraperitoneal injection of ketamine (100 mg/Kg) and xylazine (10 mg/Kg). In total, 6 µL of FITC-Dextran were injected into the footpad of the lower right limb. The skin was carefully removed to expose afferent collecting lymphatic vessels to the PLN. The mouse was then placed into a petri dish and onto the stage of an inverted microscope (Leica, DMi8). Four videos of 90 s were acquired per mice and the number of contraction and dilatation (difference between maximum and minimum diameter) was analyzed using Fiji software. To assess the effect of APLN, lentiviral vector was injected into the derma in the lower right limb 7 days before the experiment. For L-NAME study, mice were allowed to drink freely for 7 days.

## LentiFlash® construction, production, purification, and quantitation by p24 ELISA assay

Four plasmids were used to produce recombinant LentiFlash® particles in HEK293T cells: (i) the pLVGagPol plasmid coding the viral *gag* and *pol* genes modified to harbor the PP7-Coat Protein (PCP) within the *gag* gene and referred to as pLF-GagPol ΔZF2_PCP (Mianne et al, 2022); (ii) the pVSVG plasmid coding the VSV-G glycoprotein; and (iii) the two plasmids coding each one RNA cargo, flanked by the PP7 bacteriophage aptamers to enable RNA mobilization into lentiviral particles through the interaction with the PP7 coat protein cloned in the Gag sequence. All newly generated constructs were verified by restriction enzyme digestion and sequencing. LentiFlash® particles were produced in a 10-layer CellSTACK chamber (6360cm2, Corning) after transfection of in HEK293T cells with the four plasmids using the standard calcium phosphate procedure. Twenty-four hrs post-transfection, the supernatant was discarded and replaced by fresh medium, and cells were incubated at 37 °C in a humidified atmosphere of 5% $CO_2$ in air. After medium change, supernatant was collected, clarified by centrifugation at 3000× $g$ for 5 min, and microfiltered through 0.45-µm pore size sterile filter units (Stericup, Millipore). The supernatant was harvested several times, and finally all samples were pooled (crude harvest). The crude harvest was concentrated and purified by ultrafiltration and diafiltration. For quantification, the p24 core antigen was detected directly in

the viral supernatant with a HIV-1 p24 ELISA kit (Perkin Elmer), as specified by the supplier. The viral titer (expressed in physical particles per mL) was calculated from the p24 amount, knowing that 1 pg of p24 corresponds to $10E + 4$ physical particles.

## Enzyme immunoassay (EIA)

The concentration of APLN in the medium and in mouse plasma was determined using APLN EIA Kit (RAB0018, Sigma-Aldrich) using the manufacturer's recommendations. VEGF-C ELISA kit was from R&D systems.

## Cell culture and treatment

Human dermal lymphatic endothelial cells (HDLEC) (single donor, juvenile foreskin, Promocell, C-12216, >95% of the cells are CD31-positive and podoplanin positive) were cultured in Endothelial Cell Media MV2 (EGM-MV2, Promocell, C-22121). NIH3T3 were cultured in Dulbecco's modified Eagle's medium (DMEM, Sigma, D6429) supplemented with 10% fetal bovine serum (FBS, Gibco, 10270-06) and 1% penicillin–streptomycin. Endothelial cells were used at passages 3–6. Cells were cultured at 37 °C in a 5% $CO_2$ incubator. Culture medium was changed three times a week, and the cells were passaged 1/3. For collecting conditioned media, NIH3T3 was grown in 10-cm dishes, after reaching confluence, medium was removed and the cells were washed once with PBS. NIH3T3 were cultured overnight in 5 mL reduced serum medium optiMEM (Gibco), the medium was then collected and used for experiments. HDLEC were treated with 50% conditioned media/50% MV2-05% FBS. Cells are kept at low passage number (< 6) and checked for mycoplasma.

## RNA extraction, reverse transcriptase, and qPCR

Total RNA was prepared using RNeasy kit (Qiagen 74106) according to the supplier's instruction. 1 µg of RNA was retro-transcribed using high-capacity cDNA Reverse Transcription Kit (Thermo Fisher Scientific, 4368813) containing Multi-Scribe Reverse Transcriptase according to the supplier's instruction. Quantitative real-time PCR was performed using OneGreen FAST qPCR premix (Ozyme, OZYA008) on a StepOne Real-time PCR System (Thermofisher Scientific). All samples were analyzed in duplicates. Data were normalized relative to HPRT mRNA levels.

The list of primers is the following: THSB2: F—AGAGTCACTT-CAGGGGTTTGC; R—TGGCAACCCTTCTTGCTTAGA, CCBE1: F—ACATGGTGAAAGCCGGAACT; R—TTGTTGGGGAGCAG AGCAAT, E2F8: F—CATGCTCGAGGACAGTGGTT; R—GCAC TGCGTGAGAGGGATTA, APLN: F—GTTTGTGGAGTGCCAC TG; R—CGAAGTTCTGGGCTTCAC, ADAMTS3: F—TTCCA GGAACCTCTGTTGCC; R—GCTGATCTCTTGTAGACAAC, FBN: F—ACCTCA ACAGATGGCTCTCG, R—GCAGCACTG-CATTTT CGTCA, COL1A1: F—TGATGGGATTCCCTGGACCT; R-CCAGCCTCTCCATCTTTGC, HPRT: F—TGGCCATCTGCC-TAGTAAAGC; R— GGACGCAGCAACTGACATTTC, E2F1: F—AGGAACCGCCGCCGTTGTTCCCGT; R—CTGCCTGCAAA GTCCCGGCCACTT, Leptin: F—GCTGTGCCCATCCAAAAAG TCC; R—CCCAGGAATGAAGTCCAAACCG, Adiponectin: F—GTGAGAAAGGAGATCCAGGTCTT; R—TTTCCTGCCTTG GATTCCCG.

## Immunoblotting

Cells were scrapped and lysed in RIPA buffer (RIPA 2X, Biotech RB4476) supplemented with phosphatase inhibitor (PhosSTOP Easy-pack, Roche0490687001) and protease inhibitors (protease inhibitor cocktail, Sigma-Aldrich). Lysates were centrifuged at $13,500 \times g$ for 10 min at 4 °C. Supernatants were then collected and mix with Laemmli buffer containing dithiothreitol (DTT 1 mM). Proteins were resolved on 4–15% SDS-PAGE gels and transferred to nitrocellulose membrane (Trans-Blot Turbo RTA transfer kit, #1704271, Biorad). Membranes were blocked for 1 h at room temperature in 5% BSA-TBS-T (TBS-0.1% Tween 20) and probed with primary antibody overnight at 4 °C. Antibody used are the following:

Phospho-AKT: AKT-p ser473 (CS#4060S, 1/1000), AKT: Santa Cruz H136 (S8312, 1/50), Phospho ERK: ERK1/2-p(MAPKp42/44) (Thr202/Thr204) (Cell Signalling #9106, 1/2000), ERK ERK1/2-(MAPKp42/44) (Thr202/Thr204) (Cell Signalling #9102, 1/2000), Phospho-eNOS (Cell Signalling #9571S), eNOS (Cell Signalling #5880S, 1/10,000), E2F8 (Abcam, AB109596, 1/2000), VEGFR3 (R&D System AF349, 1/200), Phospho-VEGFR3 (Affinity AF3676, 1/1000), CCBE1 (Sigma SAB1402017, 1/100).

After three washes in TBS-T, membranes were probed with HRP-conjugated secondary antibodies at 1/10,000 dilution. Signals were visualized with a chemiluminescence detection reagent (Sigma) on a Chemidoc (Biorad) digital acquisition system.

## Bulk RNA sequencing

### RNA sequencing on primary human lymphatic endothelial cells

Total RNA from HDLEC treated or not with conditioned media containing APLN were harvested 24 h after treatment and isolated using the RNeasy mini kit (Qiagen). DNA digestion was performed using the RNase-Free DNase set (Qiagen). Total RNA was then subjected to ribosomal-RNA-depleted RNA sequencing (RNASeq) protocol performed by Genewiz company using Illumina HiSeq, PE $2 \times 150$ configuration.

Sequence reads were trimmed to remove possible adapter sequences and nucleotides with poor quality using Trimmomatic v.0.36. Reads were then mapped to the Homo sapiens reference genome (GRCh38) using the STAR aligner v.2.5.2b. Gene hit counts were calculated by using feature Counts from the Subread

package v.1.5.2. The hit counts were summarized and reported using the gene_id feature in annotation file. Only unique reads that fell within exon regions were counted.

### Differential gene expression analysis

After the extraction of gene hit counts, the gene hit counts table was used for downstream differential expression analysis. Using DESeq2, a comparison of gene expression between APLN-treated HDLEC against control HDLEC was performed. The Wald test was used to generate $P$ values and log2 fold changes. Genes with log2FC > 0.5 or log2FC < − 0.5 and an adjusted $P$ value < 0.05 were defined as differentially expressed genes and used for the downstream analysis. The global transcriptional change across the two groups compared was visualized by a volcano plot.

### Gene ontology (GO) analysis

A gene ontology analysis was performed separately on the statistically significant sets of upregulated and downregulated genes, using PANTHER software (version 16.0, http://pantherdb.org/). The Homo Sapiens reference list was used to cluster the set of significantly differentially expressed genes based on their biological processes or pathways and the overrepresentation of gene ontology terms was tested using Fisher exact test. All the GO terms with a False discovery rate (FDR) lower than 0.05 were considered significant and are listed in Supplementary Data.

## Chromatin immunoprecipitation (ChIP)

Human dermal lymphatic endothelial cells (HDLEC) nontrans-duced or transduced with APLN Lentivector, were crosslinked for 15 min using 1% formaldehyde directly in the culture medium. 0.125 M of Glycine were then added for 5 min. After two washes with cold PBS, cells were scraped and frozen at −80 °C. Cells were lysed to ChIP-IT Express Magnetic Chromatin Immunoprecipita-tion kit (Active Motif 53008). Optimal sonication conditions were determined previously in order to obtain DNA fragments of about 500 bp. Cells were sonicated in 350 µl final volume of Shearing buffer specific to the kit, using Diagenode Bioruptor Sonicator (7 cycles, 30 s ON, 30 s OFF in a water bath). DNA concentration was determined using a Nanodrop and 25 µg of chromatin were used by reaction. Experiments were then performed according to the manufacturer's protocol. 4 µg of E2F8 antibody (Abcam, AB109596) were used by ChIP reaction. In total, 10 µl of each sample were kept as Input. Reactions were incubated overnight at 4 °C. A mock sample without antibody was processed similarly. Prior to qPCR, DNA was purified using the Active Motif Chromatin IP DNA purification kit (58002), and eluted in 50 µl of DNase/RNase-free water. Overall, 2 µl of purified chromatin was used for qPCR. In some experiments, ChIP reactions were supplemented with 10 ng of Drosophila melanogaster chromatin (spike in chromatin, Active motif, 08221011), and 1 µg of an antibody recognizing H2Av, a Drosophila specific histone variant, (spike in antibody, active motif, 61686), as an internal control for ChIP normalization. The primers used for qPCR are the following: CCBE1: F—CCTCCTCCGTTTTCTTGTT; R— TTGTCCTGAGCGGCTT-TAAT, E2F1: F—AGGAACCGCCGCCGTTGTTCCCGT; R—TGCCTGCAAAGTCCCGGCCACTT.

**The paper explained**

**Problem**

Secondary lymphedema (LD) is a chronic condition that affects millions of cancer survivors. It is the consequence of a severe lymphatic dysfunction leading to the accumulation of fluid and fibrotic adipose tissue (AT) in a limb. It is a characterized by tortuous lymphatic capillary network and a severe decrease in lymphatic collecting vessels pumping. Whereas the vascular endothelial growth factor-C (VEGF-C) is well-known to promote lymphatic capillary growth, little is known about the regenerative control of the lymphatic collecting vessels.

**Results**

Our study addresses the role of Apelin (APLN), a key bioactive peptide, in lymphatic endothelial cells (LECs) function. We identified the loss of APLN in human LD tissue biopsies. Our study shows that APLN knockout mice recapitulate the increase of edema associated with dermal lymph backflow and skin fibrosis. In particular, lymphatic collector pumping largely depends on the APLN-mediated eNOS pathway. Overall, our data show that APLN restores collector function and participates in capillary homeostasis by stimulating VEGF-C maturation.

**Impact**

Our study reveals that targeting APLN and VEGF-C might prove useful as novel therapeutical strategy to treat the entire limb lymphatic network in LD patients. Importantly, we propose to use a nonintegrative and transient mRNA delivery strategy for patients who develop LD after cancer treatment.

## Cell transduction

NIH3T3 were seeded in six-well plates at 100,000 cells/well. After 24 h, cells were transduced with 1 mL of APLN lentivector diluted in 1 mL of OptiMEM Media in the presence of proteamine sulfate at a final concentration of 5 µg/mL. The control cells (NT) nontransduced with the APLN-, VEGF-C-, APLN + VEGF-C-lentivectors were processed at the same time and following the same protocol; in this case, 2 mL of OptiMEM and 5 µg/mL of protamine sulphate were added to the cells. The media was replaced after 24 h. Cells were grown to reach confluence, and then passed and amplified for further experiments. APLN transduction was verified by RT-qPCR.

## Statistical analysis

All results presented in this study are representative of at least three independent experiments. In all figures, "*n*" represents the number of biological replicates. Data are shown as the mean ± standard error of the mean (s.e.m.). Statistical significance was determined by two-tailed Student's *t* test, one-way ANOVA or two-way ANOVA test with Bonferroni post hoc test using Prism ver. 9.0 (GraphPad). Differences were considered statistically significant with a *P* value < 0.05. Symbols used are: ns >0.05, *≤0.05, **≤0.01, ***≤0.0001.

## Ear permeability assay

Mice were injected intradermally in the ear with 2 µL of control or APLN lentivector ($6.5.10^5$ TU/mL). After 24 h, 10 µL of Evans blue dye was injected intravenously for 10 min. Mice were sacrificed and ears were harvested and weighed before overnight incubation in

formamide at 55 °C. Then, spectrophotometric measurement (600 nm) was performed to evaluate dermal leakage.

## Data availability

The datasets produced in this study are available in the following databases: [RNAseq data]: E-MTAB-13381.

## Peer review information

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

## Acknowledgements

We thank the module of the Clinical Investigation Center (CIC 1436) from Toulouse University Hospital (Toulouse, France), in particular M. Lebrin for her outstanding assistance. We thank E Lhullier and C Segura (GenoToul platform) for their technical support as well as M. Rousseau from the platform Anexplo Genotoul (Inserm US006, Toulouse, France) for their technical advices. We thank the imaging and histology platform of I2MC institute (R Flores and C Segura) as well as genomic platform (E Lhuillier). We thank Sophie Allart, Simon Lachambre and Lhorane Lobjois for technical assistance at the cellular imaging facility of Inserm UMR 1291, Toulouse. This work has received funding from the European Union's Horizon 2020 research and innovation program named Theralymph under grant agreement no. 874708. This work has been supported by the Foundation for Medical Research (FRM).

## Author contributions

**Justine Creff**: Conceptualization; Investigation. **Asalaa Lamaa**: Conceptualization; Investigation. **Emeline Benuzzi**: Investigation. **Elisa Balzan**: Investigation. **Francoise Pujol**: Investigation. **Tangra Draia-Nicolau**: Software. **Manon Nougué**: Investigation. **Lena Verdu**: Investigation. **Florent Morfoisse**: Supervision; Investigation. **Eric Lacazette**: Validation. **Philippe Valet**:

Validation; Methodology. **Benoit Chaput**: Investigation. **Fabian Gross**: Methodology. **Regis Gayon**: Investigation; Methodology. **Pascale Bouillé**: Supervision; Validation. **Julie Malloizel-Delaunay**: Validation. **Alessandra Bura-Rivière**: Validation. **Anne-Catherine Prats**: Supervision; Visualization; Writing —original draft; Writing—review and editing. **Barbara Garmy-Susini**: Conceptualization; Formal analysis; Supervision; Funding acquisition; Validation; Writing—original draft; Project administration; Writing—review and editing.

## Disclosure and competing interests statement

The authors declare no competing interests.

# Expanded View Figures

**Figure EV1.   Effect of APLN vector on skin angiogenesis.**

(**A**) Skin section staining with Lyve1 (green) and APJ (red). Scale bar = 50 μm. (**B**) Skin section staining with vimentin (green) and APJ (red). Scale bar = 50 μm. (**C**) Skin section staining with CD68 (green) and APJ (red). Scale bar = 50 μm. (**D**) APLN mRNA expression in skin from ctrl and LD limb. Data represent mean ± SEM ($n = 3$) (*$P < 0.05$, unpaired $t$ test). (**E**) Skin sections of control or LD limb from LV-APLN-treated mice were stained for CD31 (red). DNA was stained with DAPI. Scale bar = 100 μm. (**F**) Quantification of blood vessels per field according the limb and treatment of mice. Data represent mean ± SEM ($n = 9$ control and $n = 8$ APLN-treated mice) (*$P < 0.05$, **$P < 0.01$, one-way ANOVA). (**G**) Vascular permeability assay by Evans blue extravasation. Left ear intradermally injected with 2 μL of control lentivector, right ear injected with 2 μL of APLN lentivector. (**H**) Quantification of extravasated Evans blue dye. Data represent mean ± SEM (*$P < 0.05$, unpaired $t$ test).

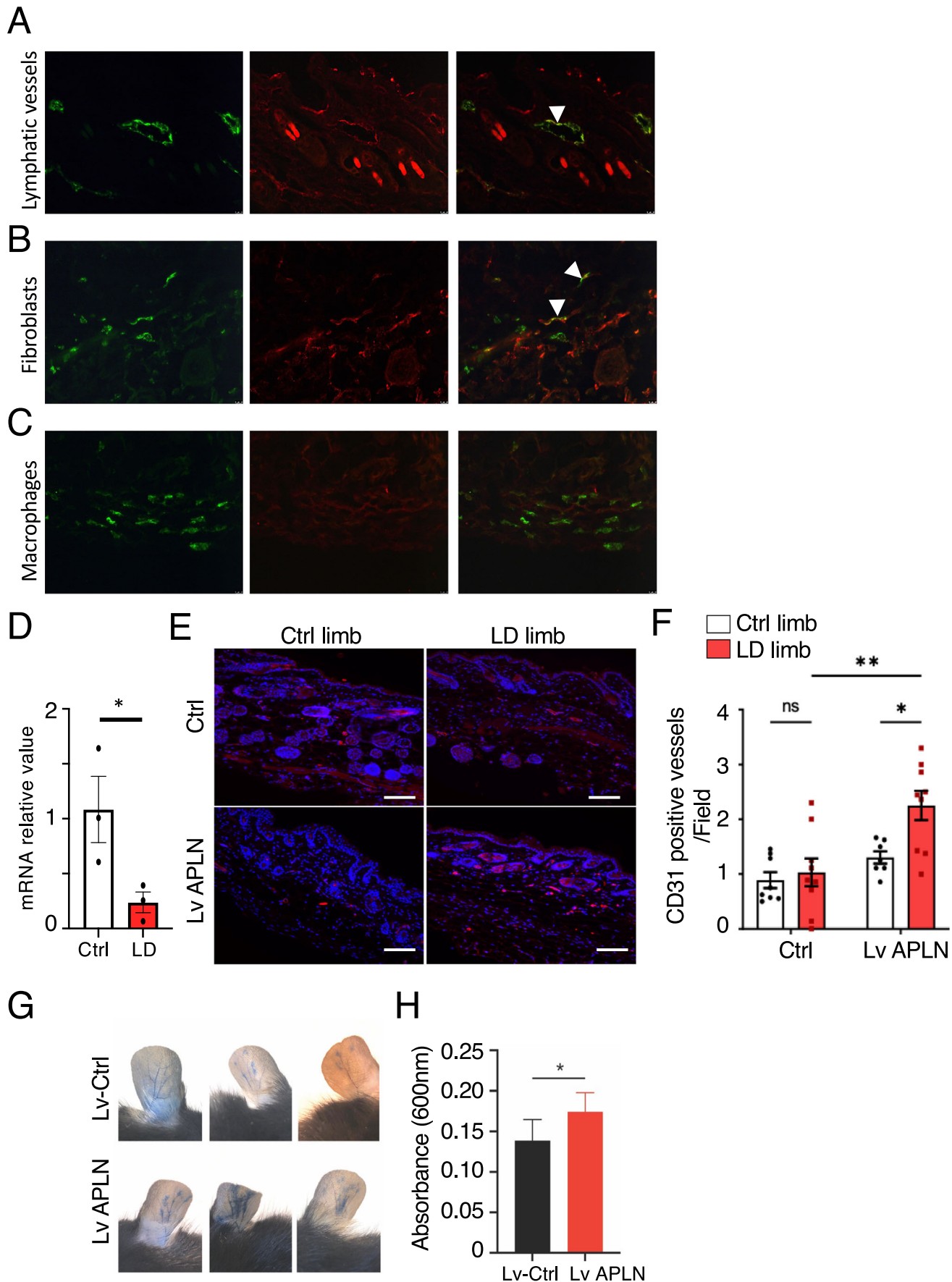

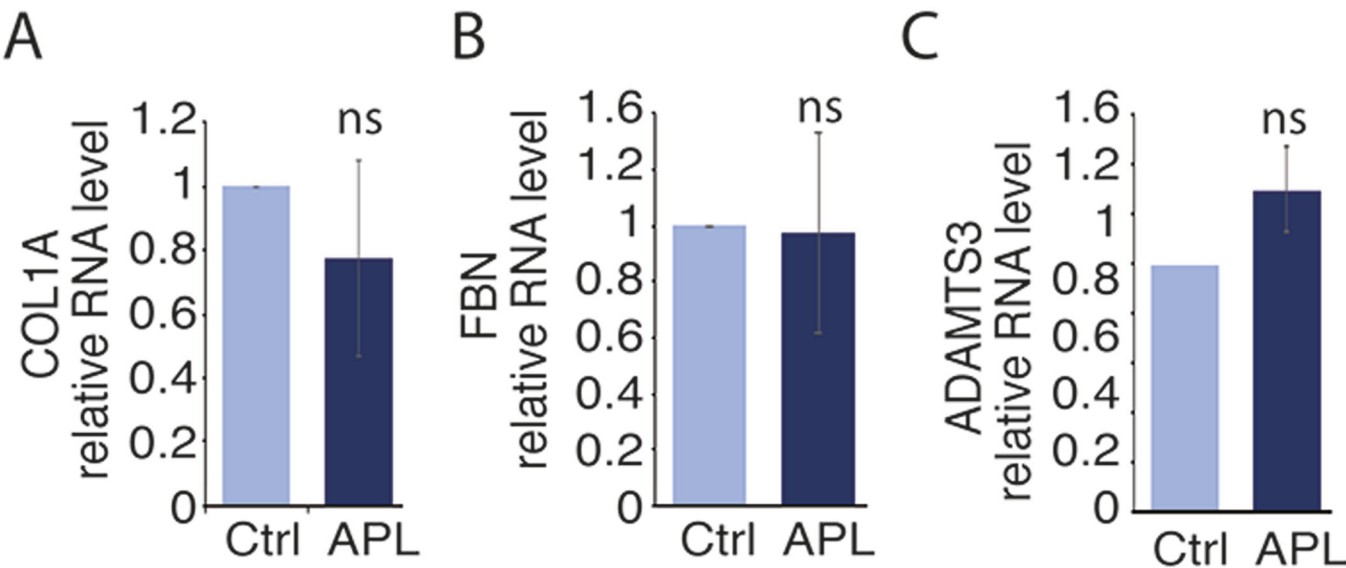

**Figure EV2. Gene expression in APLN-stimulated HDLEC.**

(**A–C**). RT-qPCR showing the expression of COL1A (**A**), FBN (**B**), and ADAMTS3 (**C**) in APLN-stimulated HDLEC. Data represent mean ± SEM (ns=non significant, unpaired *t* test).

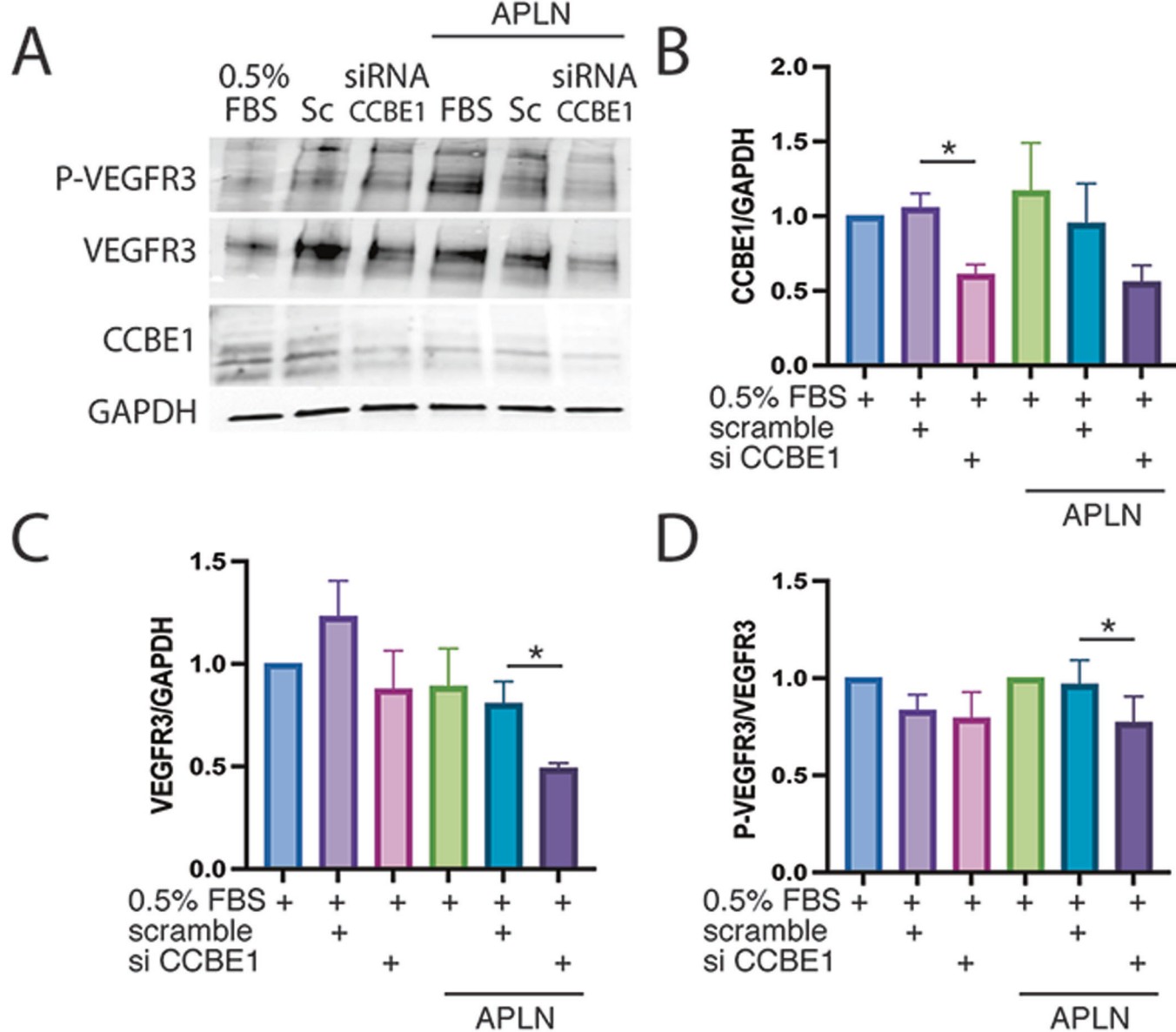

**Figure EV3. Stimulation of VEGFR3 phosphorylation by APLN.**

(A) Representatives phospho-VEGFR3/VEGFR3 and CCBE1 immunoblots of HDLEC treated with APLN +/− siRNA CCBE1. (B) Graphs represent quantification of CCBE1/GAPDH protein ratio from at least three independent experiments. Data represent mean ± SEM ($n = 3$ independent replicates) (*$P < 0.05$, two-way ANOVA). (C) Graphs represent quantification of VEGFR3/GAPDH protein ratio from at least three independent experiments. Data represent mean ± SEM ($n = 3$ independent replicates) (*$P < 0.05$, two-way ANOVA). (D) Graphs represent quantification of phospho/total protein ratio of VEGFR3 from at least three independent experiments. Data represent mean ± SEM ($n = 3$ independent replicates) (*$P < 0.05$, two-way ANOVA).

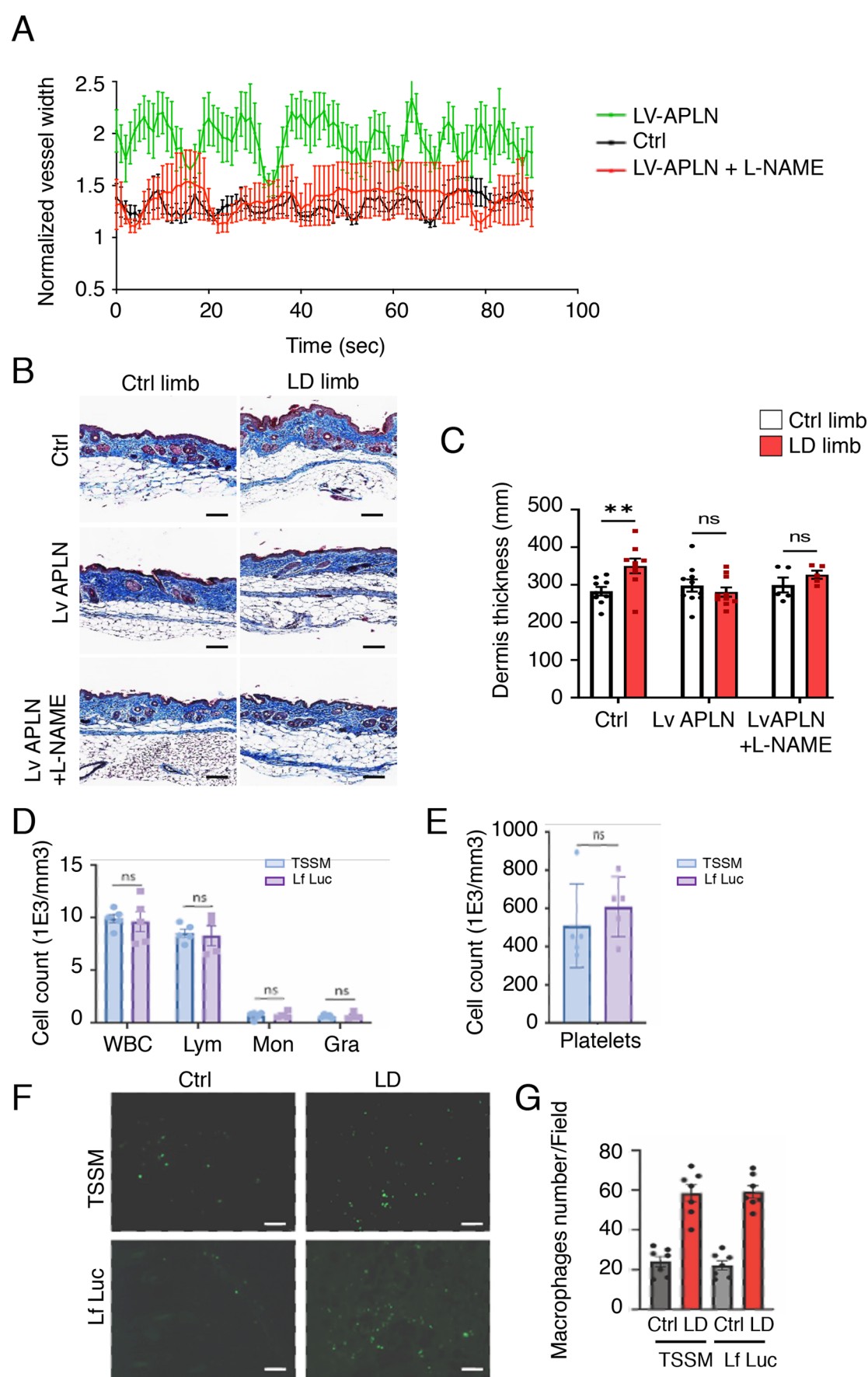

◀ **Figure EV4. Effect of L-NAME on dermis fibrosis.**

(A) Graph represent mean vessel width normalized to the minimum vessel width. (LV-APLN $n = 8$, Ctrl $n = 7$ and LV-APLN + L-NAME $n = 4$). (B) Fibrosis was evaluated by Masson's trichrome staining of the skin in control mice ($n = 9$) treated with APLN lentivector ($n = 9$) and LV-APLN + L-NAME ($n = 5$). (scale bar: 50 μm). (C) graph display the quantification of dermis thickness. Data represent mean ± SEM ($n = 9$) (**$P < 0.01$, two-way ANOVA). (D) Number of total circulating white blood cells, lymphocytes, monocytes and granulocytes in blood from mice 2 weeks after Lentiflash-APLN intradermal injection. Data represent mean ± SEM ($n = 5$) (ns = non significant, unpaired $t$ test). (E) Number of blood platelet cells from mice 2 weeks after Lentiflash-APLN intradermal injection. Data represent mean ± SEM ($n = 5$) (ns=non significant, unpaired $t$ test). (F) Representative images of macrophages (F4/80) immunodetection at the site of injection in skin from mice injected with Lf (Lf Luc) or adjuvant (TSSM) (scale bar = 50 μm). (G) Quantification of the number of macrophages at the site of Lf injection or TSSM control. Data represent mean ± SEM ($n = 7$).

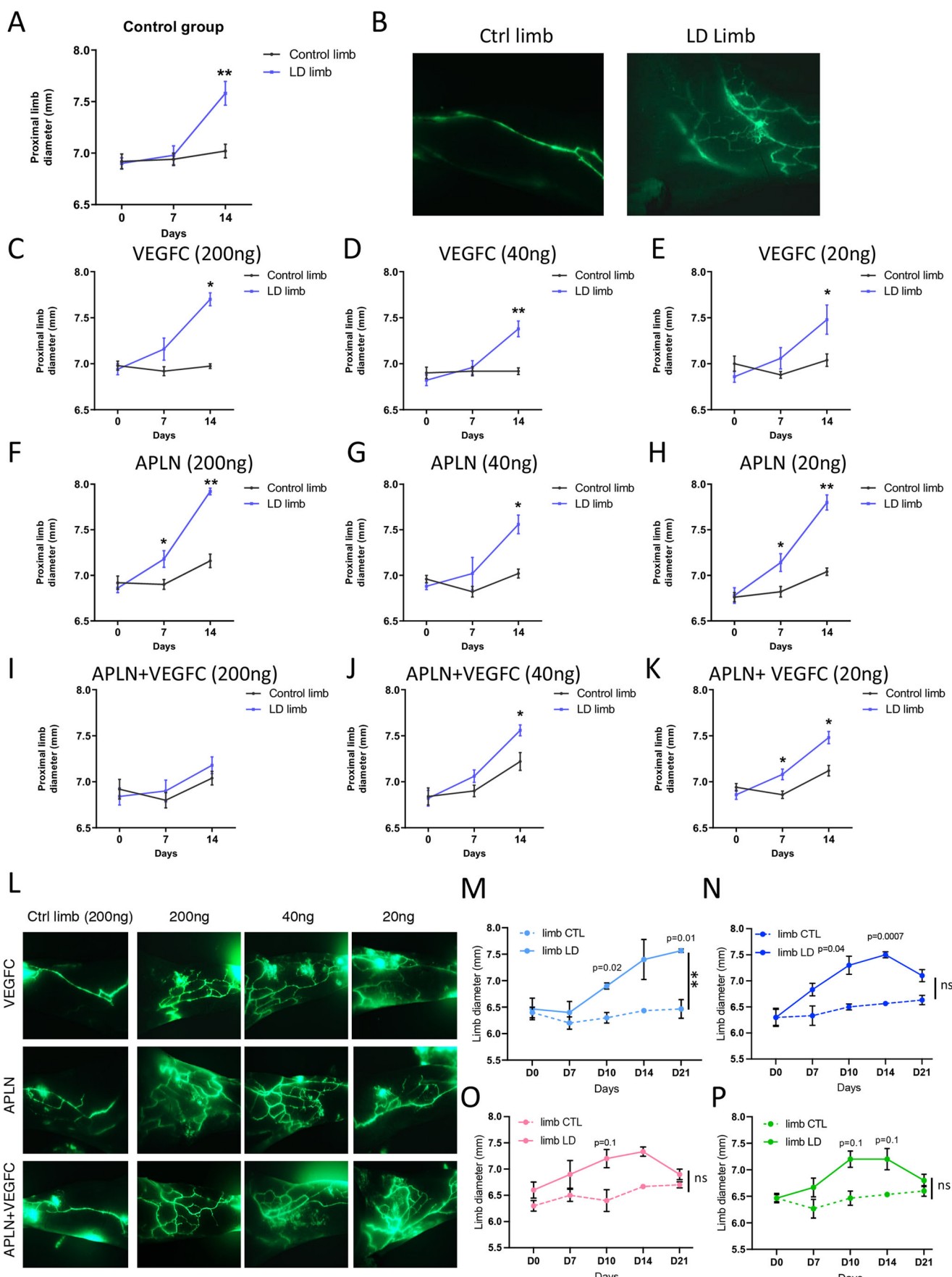

◀ **Figure EV5.  VEGF-C-, APLN, and APLN-VEGF-C Lf dose response.**

(A) Quantification of proximal limb swelling at 7 and 14 days after surgery on control limb, LD limb or LD treated with TSSM. Data represent mean ± SEM ($n = 5$) (**$P < 0.001$, two-way ANOVA). (B) Representatives images of lymphangiography from mice treated with TSSM. Scale bar: 1 mm. (C–E) Quantification of proximal limb swelling 7 and 14 days after surgery on control or LD treated with 200 ng (C), 40 ng (D), or 20 ng (E) of VEGF-C LentiFlash® vector. Data represent mean ± SEM ($n = 5$) (*$P < 0.05$, *$P < 0.001$, two-way ANOVA). (F–H) Quantification of proximal limb swelling 7 and 14 days after surgery on control or LD treated with 200 ng (F), 40 ng (G), or 20 ng (H) of APLN LentiFlash® vector. Data represent mean ± SEM ($n = 5$) (*$P < 0.05$, *$P < 0.001$, two-way ANOVA). (I–K). Quantification of proximal limb swelling 7 and 14 days after surgery on control or LD treated with 200 ng (I), 40 ng (J), or 20 ng (K) of APLN-VEGF-C LentiFlash® vector. Data represent mean ± SEM ($n = 5$) (*$P < 0.05$, *$P < 0.001$, two-way ANOVA). (L) Representatives images of lymphangiography from mice treated with 200 ng, 40 ng, or 20 ng VEGF-C-, APLN-, or APLN-VEGF-C LentiFlash® vectors. Scale bar: 1 mm. (M) Quantification of proximal limb swelling in mice treated with TSSM. Data represent mean ± SEM ($n = 5$) (**$P < 0.001$, two-way ANOVA). (N) Quantification of proximal limb swelling in mice treated with APLN-VEGF-C LentiFlash® vector Batch 1 after LD development (10 days post surgery). Data represent mean ± SEM ($n = 5$) (two-way ANOVA). (O) Quantification of proximal limb swelling in mice treated with APLN-VEGF-C LentiFlash® vector Batch 2 after LD development (10 days post surgery). Data represent mean ± SEM ($n = 5$) (two-way ANOVA). (P) Quantification of proximal limb swelling in mice treated with APLN-VEGF-C LentiFlash® vector Batch 3 after LD development (10 days post surgery). Data represent mean ± SEM ($n = 5$) (two-way ANOVA).

