## [Peer Review File · EMBO Molecular Medicine]

Apelin-VEGF-C mRNA delivery as therapeutic for the treatment of secondary lymphedema

Justine Creff, Asalaa Lamaa, Emeline Benuzzi, Elisa Balzan, Françoise Pujol, Tangra Draia-Nicolau, Manon Nougue, Lena Verdu, Florent Morfoisse, Eric Lacazette, Philippe Valet, Benoit Chaput, Fabian Gross, Régis Gayon, Pascale Bouille, Julie Malloizel-Delaunay, Alessandra Bura-Riviere, Anne-Catherine Prats, and Barbara Garmy-Susini

DOI: 10.15252/emmm.202317394

Corresponding author: Barbara Garmy-Susini (barbara.garmy-susini@inserm.fr)

Review Timeline:

Submission Date:	5th Jan 23
Editorial Decision:	1st Feb 23
Revision Received:	13th Jul 23
Editorial Decision:	15th Aug 23
Revision Received:	3rd Oct 23
Editorial Decision:	21st Nov 23
Revision Received:	6th Dec 23
Accepted:	7th Dec 23

Editor: Lise Roth

Transaction Report:

1st Feb 2023

Dear Dr. Garmy-Susini,

Thank you for the submission of your manuscript to EMBO Molecular Medicine. We have now heard back from the three referees. As you will see below, they raise substantial concerns on your work, which unfortunately preclude its publication in EMBO Molecular Medicine in its current form.

The reviewers find that the question addressed by the study is of potential interest, however they remain unconvinced that some of the major conclusions are sufficiently supported by the data. They particularly regret the lack of rigor of analysis and lack of transparency in statistics and experimental data.

Addressing the reviewers concerns in full will be necessary for further considering the manuscript in our journal except for point #3 from referee #3. The reviewers indeed agreed that addressing this point was not essential for the conclusions of this study since it is primarily focused on therapeutic use of APLN.

Revising the manuscript according to the referees' recommendations appears to require a lot of additional work and experimentation, and given the potential interest of your findings, we are ready to extend the deadline to 6 months with the understanding that acceptance of the manuscript would entail a second round of review.

EMBO Molecular Medicine encourages a single round of revision only and therefore, acceptance or rejection of the manuscript will depend on the completeness of your responses included in the next, final version of the manuscript. For this reason, and to save you from any frustrations in the end, I would strongly advise against returning an incomplete revision. Should you find that the requested revisions are not feasible within the constraints outlined here and prefer, therefore, to submit your paper elsewhere, we would welcome a message to this effect.

We require:

- 1) A .docx formatted version of the manuscript text (including legends for main figures, EV figures and tables). Please make sure that the changes are highlighted to be clearly visible.
- 2) Individual production quality figure files as .eps, .tif, .jpg (one file per figure). For guidance, download the 'Figure Guide PDF' (<https://www.embopress.org/page/journal/17574684/authorguide#figureformat>).
- 3) At EMBO Press we ask authors to provide source data for the main figures. Our source data coordinator will contact you to discuss which figure panels we would need source data for and will also provide you with helpful tips on how to upload and organize the files.
- 4) A .docx formatted letter INCLUDING the reviewers' reports and your detailed point-by-point responses to their comments. As part of the EMBO Press transparent editorial process, the point-by-point response is part of the Review Process File (RPF), which will be published alongside your paper.
- 5) A complete author checklist, which you can download from our author guidelines (<https://www.embopress.org/page/journal/17574684/authorguide#submissionofrevisions>). Please insert information in the checklist that is also reflected in the manuscript. The completed author checklist will also be part of the RPF.
- 6) Please note that all corresponding authors are required to supply an ORCID ID for their name upon submission of a revised manuscript.
- 7) It is mandatory to include a 'Data Availability' section after the Materials and Methods. Before submitting your revision, primary datasets produced in this study need to be deposited in an appropriate public database, and the accession numbers and database listed under 'Data Availability'. Please remember to provide a reviewer password if the datasets are not yet public (see <https://www.embopress.org/page/journal/17574684/authorguide#dataavailability>).

In case you have no data that requires deposition in a public database, please state so in this section. Note that the Data

Availability Section is restricted to new primary data that are part of this study.

8) For data quantification: please specify the name of the statistical test used to generate error bars and P values, the number (n) of independent experiments (specify technical or biological replicates) underlying each data point and the test used to calculate p-values in each figure legend. The figure legends should contain a basic description of n, P and the test applied. Graphs must include a description of the bars and the error bars (s.d., s.e.m.). Please provide exact p values.

13) Author contributions: CRediT has replaced the traditional author contributions section because it offers a systematic machine readable author contributions format that allows for more effective research assessment. Please remove the Authors Contributions from the manuscript and use the free text boxes beneath each contributing author's name in our system to add specific details on the author's contribution. More information is available in our guide to authors.

16) As part of the EMBO Publications transparent editorial process initiative (see our Editorial at <http://embomolmed.embopress.org/content/2/9/329>), EMBO Molecular Medicine will publish online a Review Process File (RPF) to accompany accepted manuscripts.

In the event of acceptance, this file will be published in conjunction with your paper and will include the anonymous referee reports, your point-by-point response and all pertinent correspondence relating to the manuscript. Let us know whether you agree with the publication of the RPF and as here, if you want to remove or not any figures from it prior to publication.

I look forward to receiving your revised manuscript.

Yours sincerely,

Lise Roth

***** Reviewer's comments *****

Referee #1 (Comments on Novelty/Model System for Author):

Lymphedema model in mice that is used mimics human secondary lymphedema following breast cancer surgery to the extent possible. No ethical issues.

Referee #1 (Remarks for Author):

Manuscript by Lamaa et al. investigates the function of apelin (APLN) and APLN-VEGF-C combination in treatment of secondary lymphedema (LD). The authors identified reduced apelin levels in lymphedema and hypothesized that adding apelin would help ameliorate LD in experimental model of surgery induced LD. Evidence is presented that demonstrates a role of APLN in delaying and preventing LD, and of cooperation between VEGF-C and APLN. APLN/VEGF-C combined effect on LD is particularly impressive. Data provide rationale for mRNA delivery of APLN/VEGF-C as a therapeutic for the treatment of secondary lymphedema in the future. There are some interesting insights into the mechanism of APLN action on lymphatics, such as that it promotes lymphatic contraction. Overall, this is an interesting and important work with some weaknesses that need to be addressed.

Specific points:

1. Fig. 1C: Dilated lymphatics are shown in LD - this should not be interpreted as overloaded vessels since the size of the vessel does not indicate its functionality.
2. What is the timeline of LD development in patients whose tissue samples were analyzed? How long after the surgery did LD develop and how long did these patients have lymphedema? It is important to understand whether the changes observed are acute or chronic.
3. Fig. 1C and F: Labels seem to be misplaced - podoplanin stain is shown in 1C and M' Trichrome. is in 1F. Please correct.
4. Fig. 2E: Compared to wt control, evidence for increased dermal thickness in wt LD and in APLN^{-/-} LD is not convincing, because measurements of the dermis were taken in different areas for each condition. Please address by improving quantification. Fig. 3D in comparison shows similar type of data clearly. What is clearly visible in Fig. 2E however, is increased thickness of the epithelium in wt LD, and not in any of the other conditions. Please comment.
5. Fig. 2B-F: Please clarify at what timepoint are all comparisons done? It should be at four weeks if LD in APLN^{-/-} and wt is to be compared.

6. Fig. 3G: It is unlikely that there are almost none or only few vessels in control normal skin. Therefore, it is not convincing that lymphatic densities are increased in LD. These data need to be further substantiated, and for these types of analyses images should be shown at both low and high magnification. What is considered in the analysis - dermal lymphatics only or also lymphatics in the subcutaneous tissue?

7. Fig. 4B, D, E, F: It is problematic that some of the key DESeq data could not be validated by qPCR, especially since the sample is cells in vitro. What is the number of samples in the analysis please? It should be three for each condition. Were data repeated in an independent experiment? It is OK to focus on the genes that could be validated, but there should be three independent samples in the experiment and/or two independent experiments should be done. qPCR data in Fig. 4 D, E, F showing inconsistency of the data should not be in the main manuscript.

8. Please demonstrate which cells in the skin express receptor for APLN. It is of key importance for the data interpretation to determine whether the effect of APLN on LD is a direct effect on LECs, or whether APLN also acts indirectly through one or more cell types. APLN has many functions, and it has broad cell targets, therefore the mechanism may not involve only direct action on lymphatics. While it is not expected to dissect the mechanisms potentially involving other cells, there should at least be insight if LECs are the primary cell responding here or if other cells may play a role as well.

9. Fig. 5: Data shown here does not add much since AKT pathway has already been implicated with APLN

10. Evidence for control of fibrosis by APLN is not strong.

11. Please indicate the number of samples in each experiment (n) everywhere in Figure legends.

12. The authors state that the lentiviral platform used to deliver VEGF-C/APLN will address the problem of potential negative effects of delivering VEGF-C/APLN to cancer patients, that could potentially promote metastasis. This statement can only be made if there is data demonstrating safety, which there is not. It is not expected that the effects of VEGF-C/APLN therapy on cancer progression will be addressed in this study, but the authors should be more cautious in language. Potential advantages of mRNA approach should of course be mentioned, but the statements about safety should not be made since there is no data to support it. This is a very important issue, however, that will need to be addressed in future studies.

Referee #2 (Remarks for Author):

Lamaa and colleagues present an interesting finding that delivery of Apelin RNA may alleviate lymphedema in their submitted work. Other key and interesting findings include that APE induces expression of the lymphangiogenic support factor CCBE1 in lymphatic endothelial cells and that APE also induced pAkt and regulates NO activity in lymphatic vessels. The concerns with the work are mostly minor with the interpretation of the findings and some controls potentially lacking.

It is not clear what 'synergy' exists between APE and VEGFC. It is not demonstrated really in any data, and comparisons between APE and APE+VC are lacking in Figure 8.

If APE was increasing CCBE1, and this was what drove LAG, this could be demonstrated by either measuring VC cleavage or VEGFR3 activity (pR3). Any delivery vector should likely encode the active form of VC, so it is proposed that APE increases endogenous VC activity (and it that is the idea, evidence is needed)

Since the limb volume is reduced in APE treatment, some of the other readouts such as backflow or fibrosis are expected. While nice for completeness, the mechanistic characterization as to whether these are driven by changes in blood side or interstitial forces is not explored. Most notably, does APE have an effect on blood leakiness: this would impact fluid volume and relative tissue fluid loads impact downstream pumping. A simple permeability demonstration would be good as presumably many more cell types respond to APE.

Minor

In intro, 2nd lymphedema can arise from more than just cancer treatments (and treatment alone is vague)

What does "lymphatic shape" mean?

mRNA vaccines do not "deliver" protein

Labels on 1 C-G are incorrect

Data 'normalized' should be =1... to what are they normalized to?

Referee #3 (Comments on Novelty/Model System for Author):

I have many technical concerns relating to the data. These make it very hard to judge the adequacy of the system. I think its fair to say that if used appropriately the model system would be fine.

Referee #3 (Remarks for Author):

Lamaa and colleagues use human data, mice and cell culture models to study the impact of APLN and VEGFC in lymphedema in preclinical mouse models. They start by finding that in human secondary lymphodema patients, APLN is downregulated in the lymphedemic arm. They then go on to find that APLN KO mice have reduced ability to recover from induced lymphedema (LD). They show that delivering APLN impacts drainage and vascularisation in LD their LD model. Mechanistically they claim from studies in cultured human LECs that APLN stimulates CCBE1 expression by increasing its transcription via E2F8 activity and that this leads to increased VEGFR3 signalling to ERK and AKT. in vivo this leads to increased eNOS signalling and improving lymphatic pumping. Finally, they claim that APLN-VEGFC mRNA co-treatment improves lymphatic function in mice when delivered with a Lenti system that offers up a translational possibility for patients with LD.

While the therapeutic potential of APLN may well be promising and this is an exciting and important area of research, there are many major technical concerns with experiments throughout. It is often difficult to understand how robust and reproducible many of the central observations are with the information and statistical analysis that is provided. Mechanistically, the authors present evidence that points to some feasible mechanisms, but the analyses lack sufficient depth to give confidence in what they are concluding. Overall, the presented data is unconvincing and as such the manuscript would need extensive additional work to be suitable for publication.

Below are listed several major and minor concerns.

Major:

1. The findings in Figure 8 are central to the impact of this study. However, there are confusing measurements and the presentation of the data in figure 8 makes it difficult to appreciate if it is robust and reproducible.

Firstly, the controls plotted in Figure 8D and E appear to be precisely identical control curves used multiple times over different graphs. Are these the same animals used as controls in both graphs? Is the LD control also the same animals and measurements in Figure panel 8F? If so, why is the control curve in Figure 8F the only control that is different across the 3 plots? Why was the co-treatment (the pivotal one for the biological findings presented) compared to the same LD control but a different untreated control than each of the single treatments? The authors need to clarify the reasons for the above.

Secondly, it would be appropriate to plot all of the response curves on a single plot for Figure 8D-F and to perform transparent statistical analysis as a multiple comparison; all of the treatments compared to each other as well as to controls at each time point. Isolating each treatment into separate graphs may well give a suggestion of statistical significance that is not real when all of the samples are taken into account. All of the statistical analyses used and p-values should be provided for all points on these plots. The authors should consider the appropriate statistical tests for this type of data and what to measure. A permutation test or at least multiple anova for each stage post treatment may be suitable. All individual data points (number of mice measured) should be displayed and the standard deviation indicated.

Finally, why is the proximal limb diameter at 0 days post-surgery so much lower than the LD and untreated control for the VEGFC treatment only? Were the mice not age and size matched? This should not be the case if surgeries were comparable. If the data in this treatment were normalised and the rate of recovery measured (rather than limb diameter) it would appear to the eye that the VEGFC has no impact.

As presented the analysis in Figure 8 raises a number of concerns. It is unconvincing without significant clarification and additional transparent statistical analysis. I also was unable to figure out from the figure legend, text or methods how many mice were measured at each condition and stage.

2. The controls used for the lentiviral vector treatments may not be appropriate. In the methods it states that "The day of the surgery, vehicle or APLN lentiviral vectors were injected intradermally in the LD limb". If the control was just a vehicle injection and not a control lentiviral delivery (eg. with mutated APLN, VEGFC, mock or no growth factor) then how is the impact of the Lentivirus treatment controlled? As inflammation is well established to drive lymphangiogenesis through local growth factor production, lentiviral treatments need to also be controlled and tested for induction of inflammation. The authors need to carefully control with a mock Lenti delivery rather than vehicle and examine the possibility that inflammation is driving the changes they see.

3. The fact that the APLN KO mice have lymphoedema is interesting but hard to directly relate to the human finding of down

regulated APLN. Following surgery in the WT mice, is APLN expression changes in LD as it is in humans? Further evidence supporting a robust and reproducible difference in APLN levels in LD would improve the study.

4. The authors propose that APLN up-regulates CCBE1 to then increase downstream VEGFR3 ERK AKT signalling activity. Ccbe1 is not known as a LEC expressed gene in previous studies. How well validated is the HDLEC cell line? Can the up-regulated expression be confirmed with immunofluorescence studies or alternative approaches? Furthermore, if the authors are proposing such a mechanism, they should confirm it with epistasis studies. They could easily knock down CCBE1 in APLN treated cells to show if downstream VEGFR3 activation is dependent upon their proposed APLN->CCBE1->VEGFR3 relationship. The current evidence is not sufficiently convincing.
5. Given the increase of E2F8 following APLN treatment at the level of gene expression (RNA levels in Fig 4H), Is the increased ChIP seen in Fig4J and K a reflection of a general, global increase, rather than specific change in CCBE1 regulation? Are there control loci known to be bound by E2F8 that do not change by ChIP in this experiment?
6. Figure 6B - Measurement of dilation as presented may be misleading as it is measured as the minimum and maximum from the movie (particularly for the LV-APLN example where the maximum was only visible at the beginning of the movie). From the movie, it is clear the vessel width varies quite significantly even within the same sample. Instead, please provide the vessel width over time, over 90 seconds (1 seconds per point), to show both increase in width and contraction number. The value can be normalised to the minimum vessel width in the sample.
7. Figure 6E - It is unclear from this experiment if L-NAME alone worsens the LD phenotype. Without LV-APLN, please show that the limb diameter of L-NAME + LD is not significantly higher than LD control.
8. Figure 8H - It is unclear how vessel areas were measured from Figure 7H. Was this based on histological sections? If so, please show representative images.
9. Figure 8I - The recovery effect of dual APLN-VEGFC mRNA delivery appears to be powerful. However, it is unclear whether treatment with APLN or VEGFC mRNA alone reproduces this and if APLN-VEGFC mRNA co-treatment is better when compared to APLN or VEGFC mRNA treatment alone. Please add APLN and VEGFC mRNA treatment alone controls to show that APLN-VEGFC mRNAs have synergistic effect in stimulating LD recovery when compared to VEGFC/APLN mRNA alone.
10. Figure 7I - The author claims that APLN/VEGFC co-treatment increases CCBE1 expression when compared to APLN/VEGFC treatment alone. However, this is not the case in Figure 7I. Can the authors discuss why this is the case.
11. The authors need to be consistent throughout in the presentation of data and show all data points in graphs for all figures (for example for Figure 2D and F and Fig 5F and Fig 8C).

Minor

How many patients were sampled to generate the qPCR data in Figure 1? Was it 3 patients, 3 biological replicates from 1 patient or do the q-pcr replicates represent technical replicates from a single patient, single sample. It is unclear currently.

Some statements are inaccurate in introduction. For example, "occurs after genetic mutation (primary LD) or arises after cancer treatments (secondary LD)" this is not the correct definition of LD. It can occur after any insult including cancer therapies, surgery or most commonly, parasitic infection in the third world. Please proof read for accuracy.

"However, more rigorous phase II and phase III clinical trials have failed to demonstrate that growth factors administered as single-agent gene therapies are beneficial in the patients with cardiovascular diseases." Statements like this should cite an appropriate reference to the clinical trial outcomes.

The reference from "Van Balkom et al " is a review that predates the characterisation of ccbe1 mutations. A newer review should be used given the context in the introduction.

Figure 1I - please correct the bottom legends of the graph.

Figure 2F - is there a significant increase in derm thickness between WT LD and APLN-/-

Figure 3I - is there a significant increase in lymphatic vessel area between Ctrl LD and LV-APLN LD?

Figure 5G - please provide n number for the analysis of cell junctions/actin cytoskeleton remodeling.

Figure 5H - Is there an alternative blot showing more consistent eNOS levels? Is there an overall increase in eNOS leading to increased P-eNOS upon VEGFC/APLN treatment?

Figure 6C - The authors claim that there is no major effect on contraction frequency but there is a noticeable increase in lymphatic contraction number in LV-APLN mice. Is this significant? The author should also consider the possibility that this may also contribute to increased lymphatic function.

***** Reviewer's comments *****

Referee #1 (Comments on Novelty/Model System for Author):

Lymphedema model in mice that is used mimics human secondary lymphedema following breast cancer surgery to the extent possible. No ethical issues.

Referee #1 (Remarks for Author):

Manuscript by Lamaa et al. investigates the function of apelin (APLN) and APLN-VEGF-C combination in treatment of secondary lymphedema (LD). The authors identified reduced apelin levels in lymphedema and hypothesized that adding apelin would help ameliorate LD in experimental model of surgery induced LD. Evidence is presented that demonstrates a role of APLN in delaying and preventing LD, and of cooperation between VEGF-C and APLN. APLN/VEGF-C combined effect on LD is particularly impressive. Data provide rationale for mRNA delivery of APLN/VEGF-C as a therapeutic for the treatment of secondary lymphedema in the future. There are some interesting insights into the mechanism of APLN action on lymphatics, such as that it promotes lymphatic contraction. Overall, this is an interesting and important work with some weaknesses that need to be addressed.

We thank referee 1 for his/her careful evaluation of our work. We appreciate the comment on the importance of finding therapeutic solution for secondary lymphedema.

Specific points:

1. Fig. 1C: Dilated lymphatics are shown in LD - this should not be interpreted as overloaded vessels since the size of the vessel does not indicate its functionality.

We agree with reviewer 1: “dilated overloaded lymphatic vessels” was changed for “dilated lymphatic vessels”

2. What is the timeline of LD development in patients whose tissue samples were analyzed? How long after the surgery did LD develop and how long did these patients have lymphedema? It is important to understand whether the changes observed are acute or chronic.

A better timeline of LD development was added to the Methods in the “Human tissue specimen” paragraph:

“Samples were obtained from archival paraffin blocks of 16 lipodermectomy specimens, obtained from patients with secondary LD, treated at Toulouse University Hospital, France, between 2015 and 2016.

Patients with a history of 1 to 12 years LD presented stage 2 LD according to the classification of the International Society of Lymphology (ISL). Eligible patients had a history of unilateral non-metastatic breast cancer without recurrence for more than 5 years. The main clinical parameters used to diagnose a significant LD included a clinical upper limb edema with a volume difference between the upper limb affected and the other upper limb of 10% or 200 milliliters (ml).”

3. Fig. 1C and F: Labels seem to be misplaced - podoplanin stain is shown in 1C and M' Trichrome. is in 1F. Please correct.

We apologize for the mislabeling of the figure 1C and 1F, figure legend has been corrected.

4. Fig. 2E: Compared to wt control, evidence for increased dermal thickness in wt LD and in APLN^{-/-} LD is not convincing, because measurements of the dermis were taken in different areas for each condition. Please address by improving quantification. Fig. 3D in comparison shows similar type of data clearly. What is clearly visible in Fig. 2E however, is increased thickness of the epithelium in wt LD, and not in any of the other conditions. Please comment. To measure the dermis thickness, skin flaps were harvested and embedded in paraffin. 3 sections of skin were entirely scanned on nanozoomer. Quantification was performed using at least 20 measurements of the length between epidermis and hypodermis per skin flap. This was clarified in the methods.

Additionally, we performed second harmonic generation microscopy to evaluate the collagen deposition in the skin (New Fig. 2F and 2G, and Fig.3G and H).

Fig. 3D shows that in the presence of APLN lentivector, there is no increase in dermis thickness compared to control limb.

5. Fig. 2B-F: Please clarify at what timepoint are all comparisons done? It should be at four weeks if LD in APLN^{-/-} and wt is to be compared.

The time point was clarified into the main manuscript as well as into the figure legend:

“We observed a strong dermal backflow in APLN-KO mice 4 weeks after surgery (Fig. 2B).”

Figure legend:

“**B.** Lymphangiography reveals dermal backflow (white arrows) and pathological remodeling of lymphatic vessel after LD in APLN KO mice 4 weeks post-surgery.

F. Quantification of dermis fibrosis in APLN KO mice 4 weeks post-surgery. (*p<0.05).”

6. Fig. 3G: It is unlikely that there are almost none or only few vessels in control normal skin. Therefore, it is not convincing that lymphatic densities are increased in LD. These data need to be further substantiated, and for these types of analyses images should be shown at both low and high magnification. What is considered in the analysis - dermal lymphatics only or also lymphatics in the subcutaneous tissue?

Lymphatic vessel density in the dermis from upper limb was performed at 200X magnification on 3 sections per mice, 8 mice in control group and 9 mice in lymphedema group, 5 pictures per skin (120 to 135 pictures/group).

The hypodermis was not quantified due to the high number of Lyve1+ macrophages.

The lymphatic density is low in normal skin dermis (2-4 lymphatic vessels/field), but was significantly increased in lymphedema skin.

Additional informations were added in Figure legend:

H. Quantification of lymphangiogenesis in dermis from APLN-treated mice (Ctrl n=8, LD n=9, *p<0.05, **p < 0.01). **I.** Quantification of lymphatic dilatation in APLN-treated mice (Ctrl n=8, LD n=9, **p < 0.01).

7. Fig. 4B, D, E, F: It is problematic that some of the key DESeq data could not be validated by qPCR, especially since the sample is cells in vitro. What is the number of samples in the analysis please? It should be three for each condition. Were data repeated in an independent experiment? It is OK to focus on the genes that could be validated, but there should three independent samples in the experiment and/or two independent experiments should be done. qPCR data in Fig. 4 D, E, F showing inconsistency of the data should not be in the main manuscript.

We analyzed 4 independent samples for DESeq, whereas 9 samples (3 independent experiments in triplicate) were used for RT-qPCR analysis. According to the reviewer instructions, we moved data to supplementary material (Fig. EV2).

8. Please demonstrate which cells in the skin express receptor for APLN. It is of key importance for the data interpretation to determine whether the effect of APLN on LD is a direct effect on LECs, or whether APLN also acts indirectly through one or more cell types. APLN has many functions, and it has broad cell targets, therefore the mechanism may not involve only direct action on lymphatics. While it is not expected to dissect the mechanisms potentially involving other cells, there should at least be insight if LECs are the primary cell responding here or if other cells may play a role as well.

APLN receptor (APJ) immunodetection on LD skin sections was performed. APJ was expressed by lymphatic endothelium and fibroblasts (Fig. EV1A and B). However, APJ was not found on immune cells as shown using macrophage immunostaining (Fig. EV1C).

These data were added to the manuscript.

9. Fig. 5: Data shown here does not add much since AKT pathway has already been implicated with APLN

We agree we reviewer 1 as AKT pathway induced by APLN in LECs was previously described. However, the figure 5 aims at showing that APLN-induction of signaling molecules is similar to the activation induced by VEGFC in term of time point and intensity.

This is now mentioned in the manuscript: “Interestingly, HDLEC responded similarly to VEGF-C and APLN after 30 min as we observed a strong activation of AKT and ERK (Fig. 5C and D).”

10. Evidence for control of fibrosis by APLN is not strong.

Fibrosis analysis was improved using second harmonic generation microscope as shown in Figure 2 (APLN KO mice) and Figure 3 (LV APLN).

11. Please indicate the number of samples in each experiment (n) everywhere in Figure legends.

Number of experiments was added in figure legends.

12. The authors state that the lentiviral platform used to deliver VEGF-C/APLN will address the problem of potential negative effects of delivering VEGF-C/APLN to cancer patients, that could potentially promote metastasis. This statement can only be made if there is data demonstrating safety, which there is not. It is not expected that the effects of VEGF-C/APLN therapy on cancer progression will be addressed in this study, but the authors should be more cautious in language. Potential advantages of mRNA approach should of course be mentioned, but the statements about safety should not be made since there is no data to support it. This is a very important issue, however, that will need to be addressed in future studies.

We thank Reviewer 1 for mentioning this crucial point. We changed “safe” for “non integrative” as the safety results of the clinical trial will only be provided in 2025.

Referee #2 (Remarks for Author):

Lamaa and colleagues present an interesting finding that delivery of Apelin RNA may alleviate lymphedema in their submitted work. Other key and interesting findings include that APE induces expression of the lymphangiogenic support factor CCBE1 in lymphatic endothelial cells and that APE also induced pAkt and regulates NO activity in lymphatic vessels. The concerns with the work are mostly minor with the interpretation of the findings and some controls potentially lacking.

It is not clear what 'synergy' exists between APE and VEGFC. It is not demonstrated really in any data, and comparisons between APE and APE+VC are lacking in Figure 8. To clarify the synergistic effect of APLN and VEGFC we added a graph (Fig. 8G) comparing LD with APLN, VEGFC, APLN+VEGFC mRNA delivery to LD control. Only the addition of APLN and VEGFC was able to abolish LD. This was confirmed when performing lymphography (dermal backflow) in Fig 8H.

If APE was increasing CCBE1, and this was what drove LAG, this could be demonstrated by either measuring VC cleavage or VEGFR3 activity (pR3). Any delivery vector should likely encode the active form of VC, so is it proposed that APE increases endogenous VC activity (and it that is the idea, evidence is needed).

To answer this question, we performed the knock-down of CCBE1 in LEC using siRNA. Then, cells were stimulated by APLN and western blot analysis of P-VEGFR3 was performed.

We found that the knock-down of CCBE1 in LEC decreases the amount of VEGFR-3 protein. This was associated with a slight, but significant decrease of VEGFR-3 phosphorylation in the presence of APLN. (FigureEV3)

Since the limb volume is reduced in APE treatment, some of the other readouts such as backflow or fibrosis are expected. While nice for completeness, the mechanistic characterization as to whether these are driven by changes in blood side or interstitial forces is not explored. Most notably, does APE have an effect on blood leakiness: this would impact fluid volume and relative tissue fluid loads impact downstream pumping. A simple permeability demonstration would be good as presumably many more cell types respond to APE.

To analyze the role of APLN-mediated permeability, we used the Miles assay to measure vascular leakage in the skin. Mice were injected intradermally with control or APLN-lentivector for 24 hrs followed by intravenous injection of Evans blue dye to evaluate vessel leakage.

Spectrophotometric analysis of specimens showed that within APLN injection, we observed a significant increase of blood vessel permeability (Fig. EV1G and H)

Method was added in expanded material:

Ear permeability assay.

Mice were injected intradermally in the ear with 2uL of control or APLN-lentivector ($6.5 \cdot 10^5$ TU/mL). After 24 hours, 10uL of Evans blue dye was injected intravenously for 10 minutes. Mice were sacrificed and ear were harvested and weighed before overnight incubation in formamide at 55°C. Then, spectrophotometric measurement (600nm) was performed to evaluate dermal leakage.

Minor

In intro, 2nd lymphedema can arise from more than just cancer treatments (and treatment alone is vague)

We changed for “or arises after cancer treatments (chemotherapy and radiation) in western countries” as we do not mention the filariasis infection.

What does "lymphatic shape" mean?

« shape » was used in reference to blood vessels “shape” meaning a tube-like structure.

mRNA vaccines do not "deliver" protein

We of course agree with reviewer 2, Lentiflash particles deliver mRNA.

Labels on 1 C-G are incorrect

Figure 1 legend was corrected.

Data 'normalized' should be =1... to what are they normalized to? Normalization to 1 corresponds to control group from 1 experiment (at least 3 independent experiments per group). This normalization allowed us to also evaluate the fluctuations in control group.

Referee #3 (Comments on Novelty/Model System for Author):

I have many technical concerns relating to the data. These make it very hard to judge the adequacy of the system. I think its fair to say that if used appropriately the model system would be fine. Referee #3 (Remarks for Author):

Lamaa and colleagues use human data, mice and cell culture models to study the impact of APLN and VEGFC in lymphedema in preclinical mouse models. They start by finding that in human secondary lymphedema patients, APLN is downregulated in the lymphedemic arm. They then go on to find that APLN KO mice have reduced ability to recover from induced lymphedema (LD). They show that delivering APLN impacts drainage and vascularisation in LD their LD model. Mechanistically they claim from studies in cultured human LECs that APLN stimulates CCBE1 expression by increasing its transcription via E2F8 activity and that this leads to increased VEGFR3 signalling to ERK and AKT. in vivo this leads to increased eNOS signalling and improving lymphatic pumping. Finally, they claim that APLN-VEGFC mRNA co-treatment improves lymphatic function in mice when delivered with a Lenti system that offers up a translational possibility for patients with LD.

While the therapeutic potential of APLN may well be promising and this is an exciting and important area of research, there are many major technical concerns with experiments throughout. It is often difficult to understand how robust and reproducible many of the central observations are with the information and statistical analysis that is provided. Mechanistically, the authors present evidence that points to some feasible mechanisms, but the analyses lack sufficient depth to give confidence in what they are concluding. Overall, the presented data is unconvincing and as such the manuscript would need extensive additional work to be suitable for publication.

We thank reviewer 3 for highlighting the therapeutic potential of APLN for lymphedema. We have provided new experiments to convince reviewer 3 and we hope having reached his/her expectations.

Below are listed several major and minor concerns.

Major:

1. The findings in Figure 8 are central to the impact of this study. However, there are confusing measurements and the presentation of the data in figure 8 makes it difficult to appreciate if it is robust and reproducible.

Firstly, the controls plotted in Figure 8D and E appear to be precisely identical control curves used multiple times over different graphs. Are these the same animals used as controls in both graphs? Is the LD control also the same animals and measurements in Figure panel 8F? If so, why is the control curve in Figure 8F the only control that is different across the 3 plots? Why was the co-treatment (the pivotal one for the biological findings presented) compared to the

same LD control but a different untreated control than each of the single treatments? The authors need to clarify the reasons for the above.

We performed the APLN and VEGFC treatments in mice in the same experiment, only one group was used for control to reduce the total number of mice as asked by the local ethic committee for animal studies. APLN+VEGFC treatment was performed in another experiment. However, graph is representative from 1 in vivo experiment, but 3 independent experiments were performed to conclude to the biological effect.

To clarify the synergistic effect of APLN and VEGFC we added a graph (Fig. 8G) comparing LD with APLN, VEGFC, APLN+VEGFC mRNA delivery to LD control. Only the addition of APLN and VEGFC was able to abolish LD. This was confirmed when looking at dermal backflow readout in Fig 8H.

Secondly, it would be appropriate to plot all of the response curves on a single plot for Figure 8D-F and to perform transparent statistical analysis as a multiple comparison; all of the treatments compared to each other as well as to controls at each time point. Isolating each treatment into separate graphs may well give a suggestion of statistical significance that is not real when all of the samples are taken into account. All of the statistical analyses used and p-values should be provided for all points on these plots. The authors should consider the appropriate statistical tests for this type of data and what to measure. A permutation test or at least multiple anova for each stage post treatment may be suitable. All individual data points (number of mice measured) should be displayed and the standard deviation indicated. As Figure 8 line plots were difficult to make the graph readable, we rather made a new graph with all the treatments in lymphedema condition.

Statistical analysis was performed in accordance with biostatisticians from Inserm: to compare control limb to lymphedema limb on the same mice that received adjuvant or Lentiflash vector, we performed 2 way ANOVA test. We only revealed significant p-value for a better readable figure.

Finally, why is the proximal limb diameter at 0 days post-surgery so much lower than the LD and untreated control for the VEGFC treatment only? Were the mice not age and size matched? This should not be the case if surgeries were comparable. If the data in this treatment were normalised and the rate of recovery measured (rather than limb diameter) it would appear to the eye that the VEGFC has no impact.

Mice were age and size matched, there is only a difference of half millimeter. Using raw data or normalized data, VEGFC do not have any significant beneficial effect on LD using mRNA delivery. We only found an inhibition of LD when VEGFC is associated to APLN.

As presented the analysis in Figure 8 raises a number of concerns. It is unconvincing without significant clarification and additional transparent statistical analysis. I also was unable to figure out from the figure legend, text or methods how many mice were measured at each condition and stage.

Additional statistical analysis were performed, number of mice per experiment were added to all the figure legends.

2. The controls used for the lentiviral vector treatments may not be appropriate. In the methods it states that "The day of the surgery, vehicle or APLN lentiviral vectors were injected intradermally in the LD limb". If the control was just a vehicle injection and not a control lentiviral delivery (eg. with mutated APLN, VEGFC, mock or no growth factor) then how is the impact of the Lentivirus treatment controlled? As inflammation is well established

to drive lymphangiogenesis through local growth factor production, lentiviral treatments need to also be controlled and tested for induction of inflammation. The authors need to carefully control with a mock Lenti delivery rather than vehicle and examine the possibility that inflammation is driving the changes they see.

As Lentiflash particles cannot be correctly produced without encapsidation of mRNA, it is not possible to generate empty particles as a control. As a mock Lenti delivery we used luciferase-expression Lentiflash particles to evaluate the induction of inflammation related to Lentiflash particles. Complete blood count was performed. No increase in white blood cells (Lymphocytes, monocytes, granulocytes) or platelet number was observed after Lentiflash injection (Figure EV6).

3. The fact that the APLN KO mice have lymphoedema is interesting but hard to directly relate to the human finding of down regulated APLN. Following surgery in the WT mice, is APLN expression changes in LD as it is in humans? Further evidence supporting a robust and reproducible difference in APLN levels in LD would improve the study.

qPCR of APLN were performed to evaluate APLN expression in mouse LD. We found a decrease in APLN expression as observed in human LD (Fig. EV1D)

4. The authors propose that APLN up-regulates CCBE1 to then increase downstream VEGFR3 ERK AKT signalling activity. Ccbe1 is not known as a LEC expressed gene in previous studies. How well validated is the HDLEC cell line? Can the up-regulated expression be confirmed with immunofluorescence studies or alternative approaches?

In wildtype zebrafish, Ccbe1 was expressed spatially and temporally in the migration routes of lymphatic endothelial cells, but not in lymphatic endothelial cells themselves (Hogan et al. 2019). However, as shown in Figure EV3, we found Ccbe1 protein expression in human dermal lymphatic endothelial cells (HDLEC, neonatal) purchased from Lonza. Lymphatic endothelial cells were characterized in the laboratory by regular markers: Prox1, Lyve 1 and VEGFR3.

To confirm CCBE1 upregulation, LEC were incubated in presence of conditioned media from NIH3T3 cells +/- transduced with APLN. We confirmed the upregulation of CCBE1 protein expression in HDLEC.

Furthermore, if the authors are proposing such a mechanism, they should confirm it with epistasis studies. They could easily knock down CCBE1 in APLN treated cells to show if downstream VEGFR3 activation is dependent upon their proposed APLN->CCBE1->VEGFR3 relationship. The current evidence is not sufficiently convincing.

We performed the knock-down of CCBE1 in LEC using siRNA (Figure EV3A-D). Then, cells were stimulated by APLN and western blot analysis of P-VEGFR3 was performed (Figure EV3).

We found that the knock down of CCBE1 in LEC decreases the amount of VEGFR-3 protein. This was associated with a slight, but significant decrease of VEGFR-3 phosphorylation in the presence of APLN. (Figure EV3A-D)

5. Given the increase of E2F8 following APLN treatment at the level of gene expression

(RNA levels in Fig 4H), Is the increased ChIP seen in Fig4J and K a reflection of a general, global increase, rather than specific change in CCBE1 regulation? Are there control loci known to be bound by E2F8 that do not change by ChIP in this experiment?

Inactivation of *E2f7/8* down regulates CCBE1 and up-regulated Flt4. The expressional analysis in different cell lines (mesenchymal cells and endothelial cells) shows that *E2f7/8* are almost equally expressed in these cell lines, while productions of CCBE1 and FLT4 are dependent on cell types (Weijts BG, PLoS One. 2013, van Impel A, Adv Anat Embryol Cell Biol. 2014 Mendola A, Mol Syndromol. 2013). These data suggest that E2F7/8 does not function as an ON/OFF switch, but fine-tune the expression levels of CCBE1 and FLT4 to ensure the proper signal strength during lymphangiogenesis (Weijts BG, PLoS One. 2013)

We performed FLT4 analysis, we did not find any significant effect of E2F8 induced by APLN showing that this is not a global increase of expression of all the E2F8-regulated genes (Fig. 4I).

However, this result is in line with the literature that shows that E2F7/8 modulates CCBE1 and FLT4 in opposite ways.

6. Figure 6B - Measurement of dilation as presented may be misleading as it is measured as the minimum and maximum from the movie (particularly for the LV-APLN example where the maximum was only visible at the beginning of the movie). From the movie, it is clear the vessel width varies quite significantly even within the same sample. Instead, please provide the vessel width over time, over 90 seconds (1 seconds per point), to show both increase in width and contraction number. The value can be normalised to the minimum vessel width in the sample.

2 videos of 90 sec were analysed per mice and the vessel width was measured over 90 seconds (1 second per point) using Fiji Software. Values were normalized to the minimum vessel width in the movie. Graph is now provided in Figure EV4 and displays the mean normalized vessel width displaying both vasodilatation and contraction number.

7. Figure 6E - It is unclear from this experiment if L-NAME alone worsens the LD phenotype. Without LV-APLN, please show that the limb diameter of L-NAME + LD is not significantly higher than LD control.

According to the reviewer instructions, L-NAME alone measurement was added to the graph in Fig. 6E. We do not observe significant difference in limb diameter in LD in L-NAME treated mice compared to LD control.

8. It is unclear how vessel areas were measured from Figure 7H. Was this based on histological sections? If so, please show representative images.

The quantification of vessel area in figure 6H was performed on histological sections. Representative images are provided in Figure 6G.

9. Figure 8I - The recovery effect of dual APLN-VEGFC mRNA delivery appears to be powerful. However, it is unclear whether treatment with APLN or VEGFC mRNA alone reproduces this and if APLN-VEGFC mRNA co-treatment is better when compared to APLN or VEGFC mRNA treatment alone. Please add APLN and VEGFC mRNA treatment alone controls to show that APLN-VEGFC mRNAs have synergistic effect in stimulating LD recovery when compared to VEGFC/APLN mRNA alone.

Treatments with APLN and VEGFC (Day 10) alone were performed. No effect of APLN alone or VEGFC alone were observed in the mouse model of LD. For a greater clarity of the figure that already contains 8 graphs, data were not added to the main figure.

10. Figure 7I - The author claims that APLN/VEGFC co-treatment increases CCBE1 expression when compared to APLN/VEGFC treatment alone. However, this is not the case in Figure 7I. Can the authors discuss why this is the case.

We agree with reviewer 3 and clarified this point in the manuscript: “When comparing non treated-, VEGFC alone or APLN alone to APLN+VEGFC combination, we found an increase of genes necessary for collecting vessel function including connexin 37 and 47 (GJA4, GJC2) and claudin 5 (CDN5), whereas angiogenic genes were down-regulated (VEGFA, FLT1, KDR, KI67)(Fig.7H, and I). The expression of genes related to VEGFC maturation (ADAMTS2, CCBE1) was improved in VEGFC-, APLN-, and APLN-VEGFC groups compared to WT (Fig.7I).”

11. The authors need to be consistent throughout in the presentation of data and show all data points in graphs for all figures (for example for Figure 2D and F and Fig 5F and Fig 8C). All data points were added on graph for Figure 2D, 2F, 5F and 8C.

We also changed graphs in Figure 4E, G, H, and I.

Minor

How many patients were sampled to generate the qPCR data in Figure 1? Was it 3 patients, 3 biological replicates from 1 patient or do the q-pcr replicates represent technical replicates from a single patient, single sample. It is unclear currently.

Data from Figure 1 are generated from 8 patients for histology analysis and 2 biological replicates from 3 patients for qPCR. This is now clarified in figure legend.

Some statements are inaccurate in introduction. For example, "occurs after genetic mutation (primary LD) or arises after cancer treatments (secondary LD)" this is not the correct definition of LD. It can occur after any insult including cancer therapies, surgery or most commonly, parasitic infection in the third world. Please proof read for accuracy.

We changed for “or arises after cancer treatments (chemotherapy and radiation) in western countries” as we do not mention the filariasis infection.

"However, more rigorous phase II and phase III clinical trials have failed to demonstrate that growth factors administered as single-agent gene therapies are beneficial in the patients with cardiovascular diseases." Statements like this should cite an appropriate reference to the clinical trial outcomes.

This statement is issued from a review written by Dr Losordo in *Circulation Research* that combines all the gene therapy clinical trials performed with VEGF-A-expressing vectors in cardiovascular diseases.

Reference was added to the manuscript.

“Human studies of angiogenic gene therapy” Gupta R, Tongers J, Losordo DW. *Circ Res.* 2009 Oct 9;105(8):724-36. doi: 10.1161/CIRCRESAHA.109.200386. PMID: 19815827

The reference from "Van Balkom et al " is a review that predates the characterisation of ccbe1

mutations. A newer review should be used given the context in the introduction. According to Reviewer 3 instructions, reference was changed for:

. “CCBE1 in Cardiac Development and Disease. “

Bonet F, Inácio JM, Bover O, Añez SB, Belo JA. *Front Genet.* 2022 Feb 9;13:836694.

. “Proteolytic Cleavages in the VEGF Family: Generating Diversity among Angiogenic VEGFs, Essential for the Activation of Lymphangiogenic VEGFs. “

Künnapuu J, Bokharaie H, Jeltsch M. *Biology (Basel).* 2021 Feb 23;10(2):167.

Figure 1I - please correct the bottom legends of the graph.

Legend was corrected

Figure 2F - is there a significant increase in derm thickness between WT LD and APLN-/-

We thank reviewer 3 for his careful evaluation. The increase of dermis thickness between WT LD and APLN-/- is statistically significant. Star was added to the quantification graph. Additional experiments using second harmonic generation microscope were performed to show the increase of dermis fibrosis.

Figure 3I - is there a significant increase in lymphatic vessel area between Ctrl LD and LV-APLN LD?

No significant increase was found in lymphatic vessel area between Ctrl LD and LV-APLN LD

Figure 5G - please provide n number for the analysis of cell junctions/actin cytoskeleton remodeling.

All the *in vitro* experiments were performed at least 3 times in triplicate. This was added to the figure legend.

Figure 5H - Is there an alternative blot showing more consistent eNOS levels? Is there an overall increase in eNOS leading to increased P-eNOS upon VEGFC/APLN treatment?

Additional experiments were performed to study the effect of VEGFC+APLN on eNOS phosphorylation. No increase of eNOS was observed after VEGFC- or APLN. Additional blots and quantification were added in Figure 5H and I.

Figure 6C - The authors claim that there is no major effect on contraction frequency but there is a noticeable increase in lymphatic contraction number in LV-APLN mice. Is this significant? The author should also consider the possibility that this may also contribute to increased lymphatic function.

We agree with reviewer 3, there is a tendency for APLN to increase contraction number. However, this increase was not significant.

15th Aug 2023

Dear Dr. Garmy-Susini,

Thank you for the submission of your revised manuscript to EMBO Molecular Medicine, and please accept my apologies for the delay in getting back to you during this busy time of the year. We have now received the enclosed reports from two of the three initial referees. As you will see below, while referee #2 is overall satisfied with the revision, referee #3 still raises several major concerns on the work including the lack of proper controls, quantification and statistics.

We agree with this referee that these are essential points for the conclusions to be supported by the data and would therefore like to invite you to revise the manuscript further to address all the referee's remaining concerns. As EMBO Press usually encourages one single round of revisions, please be aware that this will be the last chance for you to address these issues. The revised manuscript will once again be subjected to review, and we cannot guarantee a positive outcome at this stage.

We have not heard back from referee #1 yet, but in order not to delay the process further and allow you to move forward, we prefer to make a decision with the reports at hand. We will forward the report from referee #1 as soon as we get it.

Moreover, please address the following editorial requests:

1/ Manuscript text:

- Please address the queries from our data editors in the related "Data edited MS file". Please remove the blue text, and only keep in track changes mode any new modification.
- Data Availability: this section should be placed below the Materials and Methods
- Acknowledgements: Please make sure that the information provided in the acknowledgements matches the information provided in the submission system (currently missing in the submission system: Cancéropôle GSO, the Foundation for medical research (FRM), the Fondation ARC pour la Recherche contre le Cancer, the National Institute of Cancer (Inca)).
- Author contributions: CRediT has replaced the traditional author contributions section because it offers a systematic machine-readable author contributions format that allows for more effective research assessment. Please remove the Authors Contributions from the manuscript and use the free text boxes beneath each contributing author's name in our system to add specific details on the author's contribution. More information is available in our guide to authors.
- Disclosure statement and competing interests: We updated our journal's competing interests policy in January 2022 and request authors to consider both actual and perceived competing interests. Please review the policy <https://www.embopress.org/competing-interests> and update your competing interests if necessary. Please rename the section accordingly.

2/ Figures:

- Please provide exact p values in your figures or in their legends, not a range.
- Please remove the movie legends from the manuscript, and zip them as a README.txt file together with the corresponding movie. The nomenclature and callouts are Movie EV1, etc..
- At EMBO Press we ask authors to provide source data for the main figures. Our source data coordinator will contact you to discuss which figure panels we would need source data for and will also provide you with helpful tips on how to upload and organize the files.
- Please upload tables S1-S4 as a single appendix file with a table of content.

3/ Synopsis: Every published paper now includes a 'Synopsis' to further enhance discoverability. Synopses are displayed on the journal webpage and are freely accessible to all readers. They include a short stand first (maximum of 300 characters, including space) as well as 2-5 one-sentence bullet points that summarize the paper. Please write the bullet points to summarize the key NEW findings. They should be designed to be complementary to the abstract - i.e. not repeat the same text. We encourage inclusion of key acronyms and quantitative information (maximum of 30 words / bullet point). Please use the passive voice. Please attach these in a separate file or send them by email, we will incorporate them accordingly. Please also suggest a striking image or visual abstract to illustrate your article as a PNG file 550 px wide x 300-600 px high.

4/ Please rename "The Paper Explains" as "The Paper Explained" and upload it as a separate file.

5/ As part of the EMBO Publications transparent editorial process initiative (see our Editorial at <http://embomolmed.embopress.org/content/2/9/329>), EMBO Molecular Medicine will publish online a Review Process File (RPF) to accompany accepted manuscripts.

In the event of acceptance, this file will be published in conjunction with your paper and will include the anonymous referee reports, your point-by-point response and all pertinent correspondence relating to the manuscript. Let us know whether you agree with the publication of the RPF and as here, if you want to remove or not any figures from it prior to publication. Please note that the Authors checklist will be published at the end of the RPF.

I look forward to receiving your revised manuscript.

Yours sincerely,

Lise Roth

***** Reviewer's comments *****

Referee #2 (Comments on Novelty/Model System for Author):

Well performed studies with little concern in their interpretation. Some statements may still be overstated as the other reviewers have identified, though I don't really see the 'many technical concerns' of others

Referee #2 (Remarks for Author):

As I only had relatively minor concerns with the original piece, the efforts by the authors have largely addressed those and I find the work acceptable. That said, the other reviewers are more concerned and their concerns are valid. I would thus be willing to accept should those concerns be equally addressed as to, for example, the controls.

Referee #3 (Remarks for Author):

The authors have addressed many of my concerns. However, there remain a number of issues that have not been dealt with well and require further effort.

Specifically, in response to:

Query 1. Regarding properly matched controls.

The authors state that "APLN+VEGFC treatment was performed in another experiment. However, graph is representative from 1 in vivo experiment, but 3 independent experiments were performed to conclude to the biological effect."

This is hard to understand. It makes it sound like they performed 3 independent experiments with no control arm and rather they used the control measurements from another, separate experiment. Proper controls need to have been used in every experiment at the same time as the experimental animals were manipulated and measured. Can the authors please be much clearer here - were there matched controls analysed side-by-side with the experimental animals at the same time and under the same conditions? If not, then in this reviewer's opinion, the experiment was not valid.

For the data represented in Figure 8 graphs 8E-G, I still am not convinced that the statistics here stack up. I would highly recommend asking an independent biostatistician to look over these data and the statistical tests that have been used.

On the query about why in Figure 8E there is such a difference in limb diameter at day 0, the authors state that:

"Mice were age and size matched, there is only a difference of half millimeter....." This response is flippant from the authors because a half millimetre difference on the other graphs would completely change the conclusions of the study. They have not explained why such a measurable difference is present. If this is an expected amount of variation, then how robust are the other measurements in other groups? Further controls or a clearer explanation of how the variation is accounted for is needed.

Query 2. Regarding lentiviral controls and local inflammation.

The authors have partially addressed the concern, but they have not examined inflammation at the site of injection. Measuring blood cell counts may not reveal if they are just inducing localised inflammation post-injection and this is the reason for more lymphangiogenesis. The authors should have stained tissues for macrophages and neutrophils after injection to address this issue. It is important to know if there is a general response to injection at play and contributing to the effect of their RNA injections.

Query 9. Regarding the need for VEGFC and APLN injections alone to be carefully measured.

The authors state:

"Treatments with APLN and VEGFC (Day 10) alone were performed. No effect of APLN alone or VEGFC alone were observed in the mouse model of LD. For a greater clarity of the figure that already contains 8 graphs, data were not added to the main figure "

It is unclear from the response if they have provided any new data in the paper and if so, where the graph is located. This is a very important control. Without knowing the proper and full effects of each treatment alone, how can the reader possibly judge the importance of the co-injections? If the data is in the paper the authors should make it clearer where and provide a clear discussion of the data.

***** Reviewer's comments *****

Referee #2 (Comments on Novelty/Model System for Author):

Well performed studies with little concern in their interpretation. Some statements may still be overstated as the other reviewers have identified, though I don't really see the 'many technical concerns' of others

Referee #2 (Remarks for Author):

As I only had relatively minor concerns with the original piece, the efforts by the authors have largely addressed those and I find the work acceptable. That said, the other reviewers are more concerned and their concerns are valid. I would thus be willing to accept should those concerns be equally addressed as to, for example, the controls. We thank reviewer 1 for supporting our work. We hope we have addressed any reviewer concerns regarding controls and statistical analysis.

Referee #3 (Remarks for Author):

The authors have addressed many of my concerns. However, there remain a number of issues that have not been dealt with well and require further effort.

We thank reviewer 3 the encouraging feedback towards our manuscript. We hope that we have responded to the concerns raised by reviewer 3.

Specifically, in response to:

Query 1. Regarding properly matched controls.

The authors state that "APLN+VEGFC treatment was performed in another experiment. However, graph is representative from 1 in vivo experiment, but 3 independent experiments were performed to conclude to the biological effect." This is hard to understand. It makes it sound like they performed 3 independent experiments with no control arm and rather they used the control measurements from another, separate experiment. Proper controls need to have been used in every experiment at the same time as the experimental animals were manipulated and measured. Can the authors please be much clearer here - were there matched controls analysed side-by-side with the experimental animals at the same time and under the same conditions? If not, then in this reviewer's opinion, the experiment was not valid.

We realize that the proper control related to experiments was not enough explicated in the manuscript. All the statistical analyses were added to the figure legends.

Additional information was added to the method:

"For each experiment, mice were injected by adjuvant or lentivector in both upper limbs. One limb had no surgery (Ctrl) and the other limb received lymphedema surgery (LD) to obtain unilateral LD as observed in patient. Each LD limb diameter was compared to the limb without surgery (Ctrl) on the same mouse."

In figure 8, results are provided by comparing 2 groups of mice (adjuvant vs lentivector) that correspond to 4 groups of treatment: Ctrl (no surgery + adjuvant), LD (surgery + adjuvant), Ctrl Lentivector (no surgery + Lentivector), LD Lentivector (surgery + Lentivector). According to statisticians, Two-way ANOVA grouped analysis was performed for statistical analysis. Graph are representative from 3 independent experiments (n=5 to 8 mice/group for each experiment)."

To do not make any confusion in Figure 8E message, graph was replaced by another experiment and its corresponding control.

For the data represented in Figure 8 graphs 8E-G, I still am not convinced that the statistics here stack

up. I would highly recommend asking an independent biostatistician to look over these data and the statistical tests that have been used.

All the statistical analysis were performed according to recommendations from independent biostatistician. For lymphedema measurement, Two-way ANOVA grouped analysis (Tukey) is necessary. Statistical test for each figure is now indicated in all the figure legend.

On the query about why in Figure 8E there is such a difference in limb diameter at day 0, the authors state that:

"Mice were age and size matched, there is only a difference of half millimeter....." This response is flippant from the authors because a half millimetre difference on the other graphs would completely change the conclusions of the study. They have not explained why such a measurable difference is present. If this is an expected amount of variation, then how robust are the other measurements in other groups? Further controls or a clearer explanation of how the variation is accounted for is needed.

The message of figure 8E aims at demonstrating that we observed a slight, but not significant effect of VEGFC-lentiflash vector compared to APLN-VEGFC-Lentiflash as shown in figure 8F and G.

This is a robust result as we performed the experiment with adjuvant, APLN alone, VEGFC alone and APLN-VEGF in three independent experiments. To better convince reviewer 2, additional experiments showing the effect of Lentiflash dose-response (VEGFC-, APLN, or APLN-VEGFC) are now provided in Figure EV6. Also, representative figure of the effect of APLN-VEGFC Lf vector were evaluated using 3 independent production batches is now provided in Figure EV7 and added to the manuscript:

“To evaluate the optimal dose of Lf vector, 3 doses (20ng, 40ng, 200ng) were evaluated on mouse model of LD (Fig. EV6 A-L). At the dose of 20 and 40ng, VEGFC-, APLN-, or APLN-VEGFC Lf had no effect on LD (Fig. EV6 A-L). At the dose of 200ng, APLN-VEGFC Lf vector rescued LD swelling, whereas VEGFC or APLN alone had no effect, confirming the beneficial outcome of the vector containing 2 RNAs (Fig. EV6 C-K). This was validated by lymphographies showing that only 200ng of APLN-VEGFC Lf vector was able to inhibit dermal lymph backflow (Fig. EV6L). To validate VEGFC-APLN combination, 3 additional experiments using 3 independent batches of APLN-VEGFC Lf vector were performed (Fig. EV7). We observed an inhibition of LD using the 3 independent production batches (Fig. EV7B-D).”

Query 2. Regarding lentiviral controls and local inflammation.

The authors have partially addressed the concern, but they have not examined inflammation at the site of injection. Measuring blood cell counts may not reveal if they are just inducing localised inflammation post-injection and this is the reason for more lymphangiogenesis. The authors should have stained tissues for macrophages and neutrophils after injection to address this issue. It is important to know if there is a general response to injection at play and contributing to the effect of their RNA injections.

Lymphangiogenesis is not induced by the lentivector or lentiflash injection as no lymphatic vessel formation or lymphatic remodeling was observed in the control limb (limb without surgery injected with lentivector). Also, no difference in lymphangiogenesis was observed between LD Ctrl (TSSM adjuvant) and LD injected with lentivector (VEGFC, APLN or APL-VEGFC).

However, it is important to notice that the increase of lymphangiogenesis in lymphedema limb (with or without lentivector) is associated with an increase of macrophages density as previously shown by our group in Morfousse F. et al. ATVB 2018.

Additional immunodetected of macrophages at the site of injection were performed as requested. We did not observe any difference in macrophage density after lentiflash vector injection compared to adjuvant in lymphedematous limb. Data are now provided in Figure EV5C and D.

Query 9. Regarding the need for VEGFC and APLN injections alone to be carefully measured. The authors state:

"Treatments with APLN and VEGFC (Day 10) alone were performed. No effect of APLN alone or VEGFC alone were observed in the mouse model of LD. For a greater clarity of the figure that already contains 8 graphs, data were not added to the main figure "

It is unclear from the response if they have provided any new data in the paper and if so, where the graph is located. This is a very important control. Without knowing the proper and full effects of each treatment alone, how can the reader possibly judge the importance of the co-injections? If the data is in the paper the authors should make it clearer where and provide a clear discussion of the data.

We agree that this sentence is confusing. These controls are presented in the manuscript and are shown in Figure 8D, 8E, and 8F.

However, when we compared the 4 treatments altogether on the same graph using normalized values (Figure 8G), we only presented lymphedema values to obtain a comprehensive graph.

In resume: graphs 8D-F show the effect on LD compared to control. Graph 8G compares the efficiency of treatments on LD, that is why we did not put ctrl (no surgery) in graph 8G as treatments have no effect on normal limb.

Also, all the independent experiments showing that APLN and VEGFC have no effect on lymphedema reduction compared to APLN-VEGFC are now provided in Figure EV6 using dose-response of lentiflash vector.

21st Nov 2023

Dear Dr. Garmy-Susini,

Thank you for submitting your revised manuscript, and please accept my apologies for the delay in getting back to you as we were waiting for one referee report. We have now received both referees' feedback, and as you will see below, while referee #2 is satisfied with the revisions, referee #3 raises remaining concerns on your study. We further discussed these concerns within the team and agreed that these points did not require additional experimental work. I am thus pleased to say that we will be able to accept your manuscript once the following points will be addressed:

1/ Referees' comments:

- Query 1: please address the referee's points regarding figure 8, in particular clarify the number of biological/technical replicates that were performed, and how the experiments were combined (data with different controls should not be combined). You might want to consider using the data from figure EV5.
- Query 9: please clarify the timeline for each experiment mentioned, including treatment administration. If the timelines are different in Figure 8J and EV5, we will NOT ask you to perform additional single mRNA treatment controls to compare the effect of co-treatment post-10 days LD, but instead to please mention this limitation in the text.
- Additionally, and as mentioned in a previous email, please clarify whether you used one-way/two-way ANOVA with repeated measures for experiments with over-time repeated measures, or alternatively whether you used one-way/2-way ANOVA (without repeated measures).

2/ Manuscript text:

- Please address the queries from our data editors in the related "Data edited MS file". In case you did not find this file, I attach it to this email for your convenience. The data editors worked on the previous version of the manuscript, so please follow the same guidelines for the new figures/figure panels.
- Please accept all previous changes, and only keep in track changes mode any new modification.
- Materials and methods:
 - o Human material: please include a statement confirming that informed consent was obtained from all subjects and that the experiments conformed to the principles set out in the WMA Declaration of Helsinki and the Department of Health and Human Services Belmont Report.
 - o Animals: please detail the housing and husbandry conditions.
 - o Antibodies: please provide dilutions/concentrations for all antibodies.
 - o Cells: please indicate whether the cells were tested for mycoplasma contamination.
 - o Statistics: please include a statement about blinding (even if no blinding was done), exclusion/inclusion criteria and about randomization, and correct the checklist accordingly. Please remove "this study includes no data deposited in external repositories." and only keep the data availability section.
- Acknowledgements: Please make sure that the information provided in the acknowledgements matches the information provided in the submission system (currently missing in the submission system: Cancéropôle GSO, the Foundation for medical research (FRM), the Fondation ARC pour la Recherche contre le Cancer, the National Institute of Cancer (Inca)).
- Please remove "Source data available online for this figure" from the figure legends.

3/ Checklist:

- Please fill in the section Cell Materials/authentication and mycoplasma contamination.
- Please complete the statistics section.
- Please fill in the section Ethics/authority granting ethics approval.

4/ Synopsis:

Thank you for providing a nice synopsis picture. Please resize it as a png/tiff/jpeg file 550 px wide x 300-600 px wide, and make sure that the text remains legible.

I look forward to receiving your revised manuscript.

Yours sincerely,

Lise Roth

***** Reviewer's comments *****

Referee #2 (Comments on Novelty/Model System for Author):

An excellent study and improved manuscript that is important to the field of lymphedema.

Referee #2 (Remarks for Author):

It appears that the authors have now included the appropriate controls as requested.

Referee #3 (Comments on Novelty/Model System for Author):

The finding is novel with clear medical impact. However, response from authors does not increase my confidence in the results.

Referee #3 (Remarks for Author):

The authors have addressed some of my comments. However, response for two key queries were very confusing, often either avoiding to address them directly or not addressing at all.

Query 1.

The authors have provided some additional information regarding how the experiments were done. However, it is still unclear why Figures 8D and 8F have identical controls (this is not internal control but LD controls), while Figure 8E have a separate one. These results are all combined in Figure 8G. This reviewer appreciate that EV5 show independent replicates of the same data (EV6/EV7 in authors response) However how the data is acquired in the main figure is very confusing. Please clarify why the controls are different for some and same for the other (D,F), how they are combined in G, and make this clear in the figure legends/methods. As of now, the data in EV5 looks clearer then the data on main Figure 8.

Query 9.

The authors have highlighted EV5 (EV6) as the single treatment controls for Figure 8J. However, all data n EV5A-K are not treated at 10 days, when LD is already present like in Figure 8J. These are not comparable and this reviewer still requires single mRNA treatment controls to compare the effect of co-treatment on reducing post-10 days LD.

1/ Referees' comments:

- Query 1: please address the referee's points regarding figure 8, in particular clarify the number of biological/technical replicates that were performed, and how the experiments were combined (data with different controls should not be combined). You might want to consider using the data from figure EV5.

The concern regarding the figure 8 is addressed in the response to reviewers below.

Number of biological replicates and respective controls are now provided in the figure legend. Here we have the same group of control for 8D and 8F, because both group were treated the same day. We have another group of control mice for 8E because it was another set of experiments.

According to the 3R: Replace, Reduce, Refine recommendations from EU, we reduce as much as possible the number of control group.

- Query 9: please clarify the timeline for each experiment mentioned, including treatment administration. If the timelines are different in Figure 8J and EV5, we will NOT ask you to perform additional single mRNA treatment controls to compare the effect of co-treatment post-10 days LD, but instead to please mention this limitation in the text.

Timeline for each experiment is indicated in Figure 8: an arrow shows the time of injection in Figure 8A (time of surgery) and in Figure 8J (10 days post-surgery).

Figure 8J and EV5 correspond to the same timeline: vector injection 10days post-surgery.

This is indicated in the figure 8J and EV5 M-P figure legend.

- Additionally, and as mentioned in a previous email, please clarify whether you used one-way/two-way ANOVA with repeated measures for experiments with over-time repeated measures, or alternatively whether you used one-way/2-way ANOVA (without repeated measures).

All the statistical analysis are now mentioned in the figure legend. 2-way ANOVA was used for comparing lymphedema.

2/ Manuscript text:

- Please address the queries from our data editors in the related "Data edited MS file". In case you did not find this file, I attach it to this email for your convenience. The data editors worked on the previous version of the manuscript, so please follow the same guidelines for the new figures/figure panels.

All the queries from data editors have been completed.

- Please accept all previous changes, and only keep in track changes mode any new modification.

Done.

- Materials and methods:

o Human material: please include a statement confirming that informed consent was obtained from all subjects and that the experiments conformed to the principles set out in the WMA Declaration of Helsinki and the Department of Health and Human Services Belmont Report.

Done

o Animals: please detail the housing and husbandry conditions.

Details were added to the manuscript in the methods section: “Mice were held under specific pathogen-free conditions in the animal facility of the Inserm Crefre US06 on a 12:12 light: dark cycle. In all cases, experimental and control animals were of the same age and gender.”

o Antibodies: please provide dilutions/concentrations for all antibodies.

Done.

o Cells: please indicate whether the cells were tested for mycoplasma contamination.

Done.

o Statistics: please include a statement about blinding (even if no blinding was done), exclusion/inclusion criteria and about randomization, and correct the checklist accordingly. Please remove "this study includes no data deposited in external repositories." and only keep the data availability section.

Done.

- Acknowledgements: Please make sure that the information provided in the acknowledgements matches the information provided in the submission system (currently missing in the submission system: Cancéropôle GSO, the Foundation for medical research (FRM), the Fondation ARC pour la Recherche contre le Cancer, the National Institute of Cancer (Inca)).

Correction was performed.

- Please remove "Source data available online for this figure" from the figure legends.

done

3/ Checklist:

- Please fill in the section Cell Materials/authentication and mycoplasma contamination.

Done

- Please complete the statistics section.

Done

- Please fill in the section Ethics/authority granting ethics approval.

Done

4/ Synopsis:

Thank you for providing a nice synopsis picture. Please resize it as a png/tiff/jpeg file 550 px wide x 300-600 px wide, and make sure that the text remains legible.

Done

***** Reviewer's comments *****

Referee #2 (Comments on Novelty/Model System for Author):

An excellent study and improved manuscript that is important to the field of lymphedema.

Referee #2 (Remarks for Author):

It appears that the authors have now included the appropriate controls as requested.

We thank referee 2 for his/her support. Remark is addressed below.

Referee #3 (Comments on Novelty/Model System for Author):

The finding is novel with clear medical impact. However, response from authors does not

increase my confidence in the results.

Referee #3 (Remarks for Author):

The authors have addressed some of my comments. However, response for two key queries were very confusing, often either avoiding to address them directly or not addressing at all.

Query 1.

The authors have provided some additional information regarding how the experiments were done. However, it is still unclear why Figures 8D and 8F have identical controls (this is not internal control but LD controls), while Figure 8E have a separate one. These results are all combined in Figure 8G.

This reviewer appreciate that EV5 show independent replicates of the same data (EV6/EV7 in authors response) However how the data is acquired in the main figure is very confusing. Please clarify why the controls are different for some and same for the other (D,F), how they are combined in G, and make this clear in the figure legends/methods. As of now, the data in EV5 looks clearer than the data on main Figure 8.

Here we have the same group of control for 8D and 8F, because both group were treated the same day. We have another group of control mice for 8E because it was another set of experiments.

According to the 3R: Replace, Reduce, Refine recommendations from EU, we reduce as much as possible the number of control group.

When different treatments/experiments are performed the same day, we only have 1 group of control (non treated mice with and without surgery).

To compare all the in vivo experiments with each other, we normalized the % of increase of lymphedema limb compared to the normal limb on the same mouse for all the mice.

Query 9.

The authors have highlighted EV5 (EV6) as the single treatment controls for Figure 8J. However, all data in EV5A-K are not treated at 10 days, when LD is already present like in Figure 8J. These are not comparable and this reviewer still requires single mRNA treatment controls to compare the effect of co-treatment on reducing post-10 days LD.

As shown in EV5 C-H, VEGFC mRNA delivery alone or APLN mRNA delivery alone injected at the time of surgery do not reduced lymphedema.

In all experiments we previously performed, injecting at the time of surgery (preventive) works better than injecting after the development of the disease (curative). Therefore, there is no evidence that VEGFC alone or APLN alone will have a beneficial effect as curative treatment, when it has no effect as preventive treatment.

In addition, in accordance with the 3R (Replacement Refinement Reduction) rules from ethic committee, there is no justification to avoid the reduction rule.

Then, we have to mentioned that VEGF-C alone did not bring any curative efficiency in human lymphedema: Phase 2 clinical trial with VEGFC-adenovirus from Herantis Pharma was cancelled in 2022 due to the lack of beneficial effect.

7th Dec 2023

Dear Dr. Garmy-Susini,

Thank you for submitting your revised files. I am pleased to inform you that your manuscript is accepted for publication and is now being sent to our publisher to be included in the next available issue of EMBO Molecular Medicine!

Please note that as discussed in my previous email, I have added the following sentence in the legend of Figure 8F: "The same control group was used in 8D and 8F". Please let us know immediately if this needs to be corrected.

Congratulations on your interesting work!

With kind regards,

Lise Roth
